**COMMUNICATIONS**

# TFEB regulates lysosomal positioning by modulating TMEM55B expression and JIP4 recruitment to lysosomes

Rose Willett[1], José A. Martina [1], James P. Zewe[2], Rachel Wills[2], Gerald R.V. Hammond[2] & Rosa Puertollano[1]

Lysosomal distribution is linked to the role of lysosomes in many cellular functions, including autophagosome degradation, cholesterol homeostasis, antigen presentation, and cell invasion. Alterations in lysosomal positioning contribute to different human pathologies, such as cancer, neurodegeneration, and lysosomal storage diseases. Here we report the identification of a novel mechanism of lysosomal trafficking regulation. We found that the lysosomal transmembrane protein TMEM55B recruits JIP4 to the lysosomal surface, inducing dynein-dependent transport of lysosomes toward the microtubules minus-end. TMEM55B over-expression causes lysosomes to collapse into the cell center, whereas depletion of either TMEM55B or JIP4 results in dispersion toward the cell periphery. TMEM55B levels are transcriptionally upregulated following TFEB and TFE3 activation by starvation or cholesterol-induced lysosomal stress. TMEM55B or JIP4 depletion abolishes starvation-induced retrograde lysosomal transport and prevents autophagosome–lysosome fusion. Overall our data suggest that the TFEB/TMEM55B/JIP4 pathway coordinates lysosome movement in response to a variety of stress conditions.

[1] Cell Biology and Physiology Center, National Heart, Lung and Blood Institute, National Institutes of Health, 50 South Drive, Building 50, Room 3537, Bethesda, MD 20892, USA. [2] Department of Cell Biology, University of Pittsburgh School of Medicine, 200 Lothrop Street, Room S332 Biomedical Sciences Tower, Pittsburgh, PA 15213, USA. Rose Willett and José A. Martina contributed equally to this work. Correspondence and requests for materials should be addressed to R.P. (email: puertolr@mail.nih.gov)

In recent years, our view of lysosomes has dramatically changed. From being considered as mere degradative organelles, lysosomes are now recognized as critical regulators of cellular homeostasis and adaptation to stress[1]. Cells modulate lysosome numbers and activity in response to a variety of external and internal stimuli. The transcription factors TFEB and TFE3 exhibit the unique ability to promote expression of multiple lysosomal genes and are considered master regulators of lysosomal biogenesis[2,3]. TFEB and TFE3 also regulate expression of genes implicated in many other cellular pathways, including autophagy, immune response, mitochondrial biogenesis, unfolded protein response, and metabolic regulation, thus revealing a critical role of these transcription factors in the coordination of different cellular stress pathways[4–8].

An emerging topic of interest is how lysosomal function may be influenced by lysosomal positioning. Lysosomes are transported bi-directionally on the microtubule network by dynein and kinesin motors. In non-polarized cells, microtubule minus-ends generally localize to the perinuclear region, close to the microtubule-organizing center (MTOC), whereas the microtubule plus-ends are directed toward the cell periphery. Therefore, minus-end-directed microtubule motors, such as dynein, move lysosomes from the periphery to the cell center, while the plus-end-directed microtubule motors, kinesins, promote scattering of lysosomes throughout the cytoplasm.

Recent evidence suggests that the distribution of lysosomes within cells is regulated in response to a variety of stimuli, and alterations in this regulation may be associated with different pathologies. For example, under starvation conditions, autophagosomes and lysosomes move toward the cell center, facilitating the fusion between both organelles and degradation of the autophagosomal content[9,10]. Redistribution of lysosomes toward the cell periphery is critical for cancer growth, invasion and metastasis. In this case, anterograde transport facilitates lysosomal exocytosis, leading to the secretion of acidic hydrolases and metalloproteinases that degrade the extracellular matrix to promote migration and invasion of cancer cells[11–14]. Lysosomal positioning also plays an important role in the immune response. Lysosome retrograde transport toward the immunological synapse is essential for the ability of natural killer and cytotoxic T lymphocytes to kill pathogen infected cells[15], whereas kinesin-dependent tubulation of lysosomes is required for efficient MHCII-mediated antigen presentation in dendritic cells[16–18].

Several protein complexes have been implicated in the regulation of lysosomal positioning. Anterograde movement of lysosomes is regulated by the multi-subunit complex BORC, the small GTPase Arl8, and its effector SKIP, which directly interacts with the kinesin light chain, thus linking lysosomes to the plus-end-directed microtubule motor kinesin[19,20]. Alternatively, a tripartite complex between Rab7, FYCO1, and kinesin has also been shown to promote outward lysosomal movement[21,22]. Retrograde movement is regulated by Rab7 and its effector RILP. RILP interacts with the p150-glued subunit of dynactin, thus recruiting the minus-end-directed microtubule motor dynein to lysosomes[23–25]. Lysosomal transmembrane proteins may also participate in the regulation of retrograde lysosomal transport. For example, LAMP-1 and LAMP-2 promote coupling of lysosomes to dynein–dynactin[26,27], whereas ALG-2 interacts with the lysosomal transient receptor potential channel MCOLN1 in a calcium-dependent manner to recruit dynein–dynactin to lysosomes[28,29]. Finally, overexpression of Rab34, Rab36, Rabring7, or Rapsyn causes clustering of lysosomes in the perinuclear area. Rab34 and Rab36 localize to the Golgi/TGN and directly interact with RILP, mediating tethering or anchoring of lysosomes to the Golgi[30,31]. It is still unclear how Rabring7 and Rapsyn may work in combination with microtubule motors[32,33].

Here we describe a novel mechanism of lysosomal positioning regulation. We found that the lysosomal protein TMEM55B interacts and recruits the dynein adaptor JIP4 to lysosomal membranes, thus inducing retrograde transport of lysosomes along microtubules. Depletion of either TMEM55B or JIP4 causes a dramatic accumulation of lysosomes at the cell periphery without affecting the distribution of early endosomes or the Golgi apparatus. Interestingly, TMEM55B is a target of TFEB and TFE3 and its mRNA and protein levels increase following TFEB/3 activation by starvation or cholesterol accumulation in lysosomes. Therefore, we propose that, by regulating TMEM55B expression levels, TFEB and TFE3 modulate lysosomal positioning in response to nutrient deprivation and lysosomal stress.

## Results

**TMEM55B promotes lysosomal retrograde transport.** Proteomic analyses have determined that lysosomes contain dozens of transmembrane proteins[34,35]; however, the physiological function of most of them remains to be elucidated. TMEM55B, a protein previously described to exhibit a primary late endosomal/lysosomal distribution[36], is of particular interest because contains several CLEAR motifs in its promoter region, suggesting that its expression is regulated by the transcription factors TFEB and TFE3[2,3]. Accordingly, ChIP-seq analysis identified TMEM55B as a putative TFE3 target both in RAW 264.7 macrophages and MEFs[5,6]. Secondary structure prediction suggests that TMEM55B is comprised of a large cytosolic N-terminal domain (CD), two transmembrane domains (TM), and a short C-terminal cytosolic tail (Fig. 1a). To gain insight into the cellular role of TMEM55B, we expressed in ARPE-19 cells a chimera in which GFP was fused to the TMEM55B N-terminal domain. In agreement with previous reports, we found that TMEM55B co-localized with the lysosomal marker LAMP-1 (Fig. 1b, top) but not with early endosome marker hepatocyte growth factor-regulated tyrosine kinase substrate (HRS) (Fig. 1b, bottom). Interestingly, upon overexpression of GFP-TMEM55B, we observed a dramatic repositioning of lysosomes, changing from dispersed throughout the cell to tightly clustered in the cell center (Fig. 1c, top). This organelle re-positioning was specific for lysosomes, as early endosome positioning was unaffected (Fig. 1c, bottom). Additionally, this lysosome clustering was not cell-type specific as evidenced by TMEM55B-induced lysosomal repositioning in HeLa cells (Fig. 1d).

To better understand the role of TMEM55B in defining lysosome sub-cellular positioning, HeLa cells were depleted of TMEM55B by RNA interference (RNAi) and lysosome distribution was analyzed. TMEM55B RNA silencing was sufficient to induce over 80% reduction in TMEM55B mRNA and protein levels (Fig. 1e, f). Due to the heterogeneity of HeLa cell shape, a linear profile (ranging from the nuclear rim to the cell edge) was used to measure lysosome distribution (Supplementary Fig. 1b). This method of analysis was comparable to measuring lysosome distribution via a radial profile (Supplementary Fig. 1a), which is generally limited to round cells for accurate measurement. Three linear profiles were averaged per cell, and the percentage of signal intensity was divided in three categories based on distance from the nuclear rim: <5 μm, 5–15 μm, or >15 μm to cell periphery. In control RNAi-treated cells, lysosomes were broadly distributed throughout the cell, with a slight concentration in the perinuclear region (Fig. 1g, h). This slight concentration has been described as the "perinuclear cloud"[37] and is the typical lysosome distribution in non-polarized cells. In cells depleted of TMEM55B the distribution significantly shifted, showing a decrease in the concentration of LAMP-1-positive structures in the perinuclear region, concomitant with a distinct increase in the

number of lysosomes localized towards the cell periphery (Fig. 1g, h).

To further corroborate these results, we assessed lysosomal distribution by measuring cumulative intensity distribution, as recently described by Starling et al.[38]. In this case, cell images were segmented by scaling the perimeter in 10% decrements and the cumulative integrated LAMP-1 intensity (relative to the whole cell) was plotted. The use of this method confirmed that the

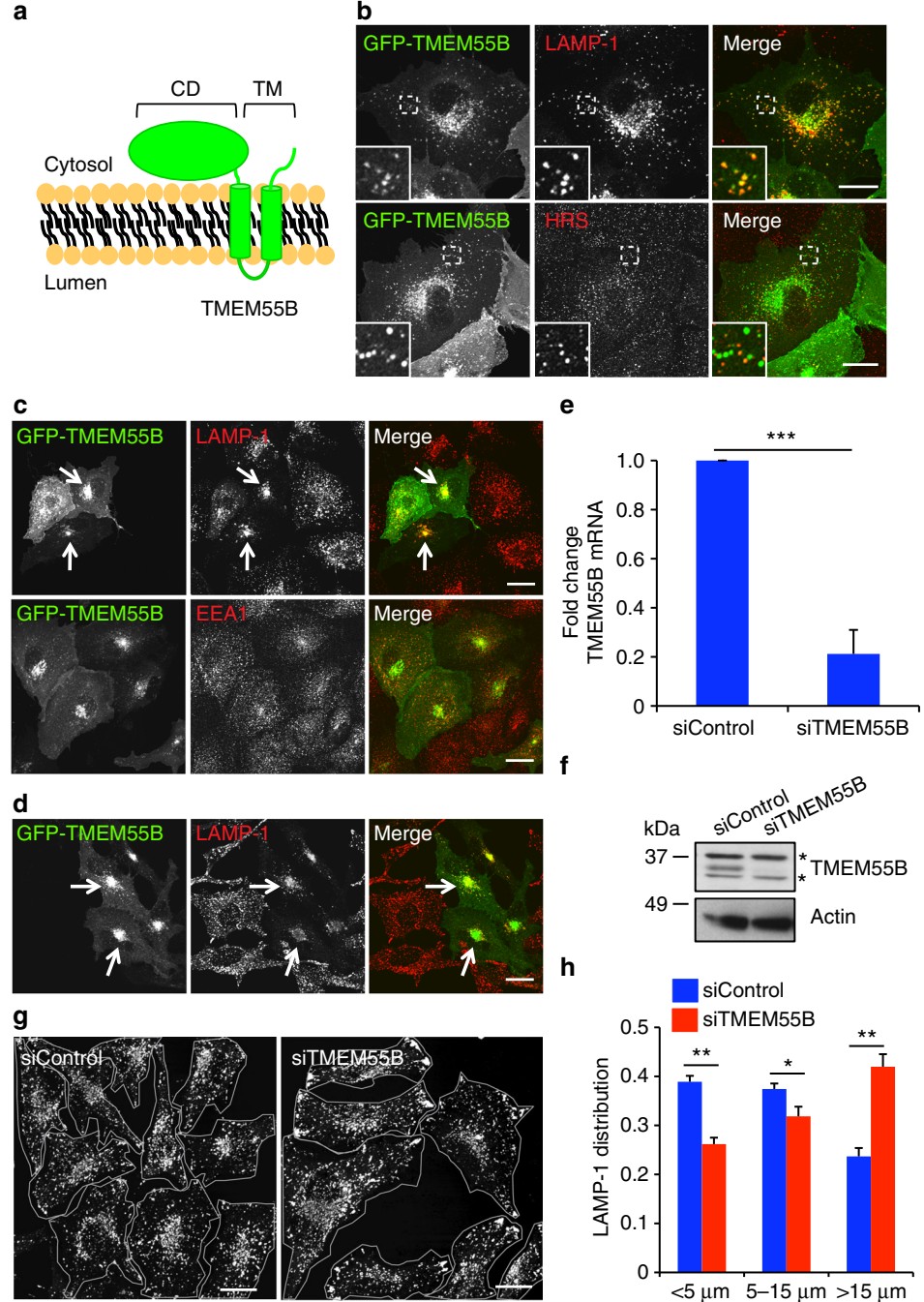

**Fig. 1** TMEM55B regulates lysosomal positioning. **a** Schematic of the predicted membrane topology of TMEM55B. **b** ARPE-19 cells transfected with GFP-TMEM55B for 18 h. Cells were fixed, permeabilized, and immunostained with antibodies against LAMP-1 (top) or HRS (bottom). Insets represent a 3.2- (top) and 4.5- (bottom) fold magnification of the indicated areas. **c** ARPE-19 cells infected with adenovirus expressing GFP-TMEM55B for 30 h. Cells were fixed, permeabilized, and immunostained with antibodies against LAMP-1 (top) or EEA1 (bottom). Arrows denote bulk of lysosomal accumulation. **d** HeLa cells infected with adenovirus expressing GFP-TMEM55B for 30 h. Cells were fixed permeabilized and immunostained with antibodies against LAMP-1. Arrows denote bulk of lysosomal accumulation. **e**, **f** HeLa cells treated with TMEM55B or Control siRNAs for 6 days. **e** Relative quantitative real-time PCR analysis of TMEM55B mRNA transcript levels (mean ± s.e.m. of the RNA fold change of indicated TMEM55B normalized to actin mRNA from three independent experiments; ***$P < 0.0001$). **f** Immunoblot of lysates of cells treated with TMEM55B or Control siRNA. **g** HeLa cells treated with TMEM55B or Control siRNA were immunostained with antibody against LAMP-1. **h** Quantification of lysosome distribution, percentage of total fluorescence signal detected at 0–5 μm, 5–15 μm, or >15 μm from nuclear rim. Quantified results are presented as mean ± s.e.m. using two-tailed $t$-test *$P < 0.05$, **$P < 0.005$ were considered significant, $n \geq 30$ from three or more independent experiments. Scale bars, 20 μm

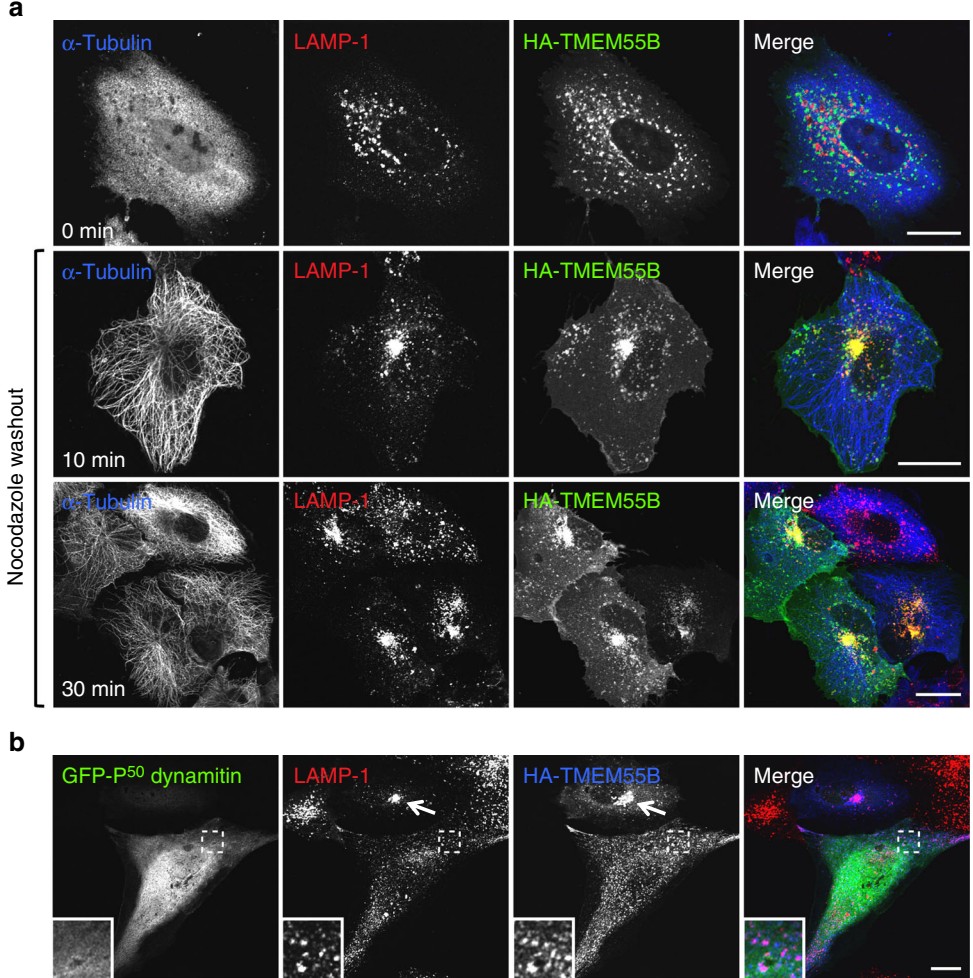

**Fig. 2** Microtubules and Dynein are required for TMEM55B-induced lysosome clustering. **a** ARPE-19 cells were transfected with 3×HA-TMEM55B and treated with 10 μM nocodazole for 2 h at 37 °C and then shifted to ice for 30 min in the presence of nocodazole. For nocodazole washouts, cells were washed and incubated in nocodazole-free culture medium for the indicated times. Cells were fixed, permeabilized, and immunostained with antibodies against LAMP-1 and HA. **b** ARPE-19 cells co-transfected with 3xHA-TMEM55B and GFP-P50 Dynamitin. Cells were fixed, permeabilized, and immunostained with antibodies against LAMP-1. Insets represent a three-fold magnification of the indicated area. Arrows indicate TMEM55B-induced lysosomal clustering. Scale bars, 20 μm

accumulation of lysosomes in the cell periphery in TMEM55B-depleted cells is highly significant (Supplementary Fig. 1c). Based on these results, we concluded that TMEM55B plays a role in determining lysosome positioning.

It is well-established that lysosomes undergo bidirectional transport along microtubule tracks[39]. Therefore, we hypothesized that TMEM55B-induced lysosomal clustering was dependent on microtubules. As expected, depolarization of microtubules with nocodazole prevented perinuclear accumulation of lysosomes in TMEM55B-expressing cells (Fig. 2a, top panel), confirming that intact microtubules were indeed required for TMEM55B-dependent lysosome movement. The block in lysosome clustering was rapidly reversed upon nocodazole washout, demonstrating the dynamic role of TMEM55B in directing lysosomes towards the cell center (Fig. 2a, bottom panels). As mentioned above, lysosomal retrograde transport is controlled by the motor dynein. Accordingly, disruption of the dynein–dynactin complex by overexpression of the dynactin complex subunit P50-dynamitin was sufficient to completely block TMEM55B-induced lysosome clustering (Fig. 2b). Thus, it was concluded that TMEM55B clusters lysosomes by recruiting the microtubule minus-end motor dynein to lysosomes, disrupting the balance of positive-

and minus-end microtubule motors directing lysosomal trafficking.

**TMEM55B CD induces organelle retrograde transport**. To better understand TMEM55B trafficking within cells, TMEM55B was divided in two domains, the N-terminal cytoplasmic domain (TMEM55B CD) and the transmembrane and C-terminal cytosolic tail domain (TMEM55B TM) (Fig. 1a). As seen in Supplementary Fig. 2a, GFP-TMEM55B CD exhibited a cytosolic distribution, whereas TMEM55B TM localized predominantly to the plasma membrane. The lack of co-localization of TMEM55B CD and TMEM55B TM with LAMP-1 was consistent with their inability to induce lysosomal clustering (Supplementary Fig. 2a). These results suggest that the full-length protein is required for proper localization to the endo-lysosomal compartment and that one or more endo/lysosomal targeting motifs likely reside within TMEM55B CD. Accordingly, alanine substitutions of Leucine10 and Leucine11, residues of a putative acidic-dileucine motif at the amino termini of TMEM55B, caused accumulation of the full-length protein at the cell surface, with a minor fraction co-localizing with LAMP-1-positive structures (Supplementary

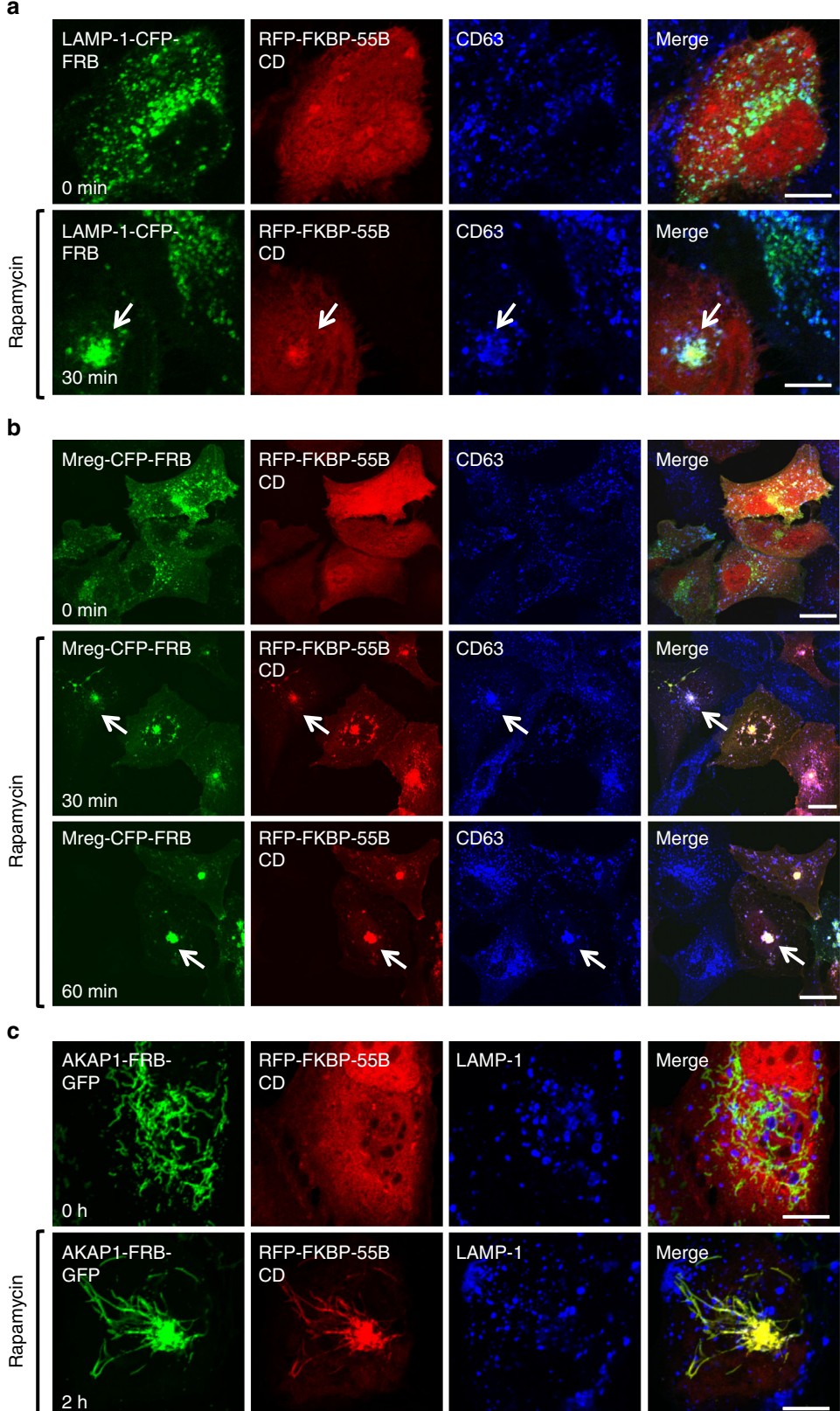

**Fig. 3** Recruitment of TMEM55B CD initiates retrograde transport of tagged membranes towards the cell center. ARPE-19 cells transfected with **a** LAMP-1-CFP-FRB and RFP-FKBP-TMEM55B CD, **b** Melanoregulin (Mreg 1-42)-CFP-FRB and RFP-FKBP-TMEM55B CD, or **c** AKAP1-FRB-GFP and RFP-FKBP-TMEM55B CD. Cells were incubated in 200 nM Rapamycin for the indicated times. Cells were fixed, permeabilized, and immunostained with antibodies against CD63 (**a**, **b**) or LAMP-1 (**c**). Arrows indicate TMEM55B CD-induced lysosomal clustering. Scale bars, 10 μm (**a**, **c**) or 20 μm (**b**)

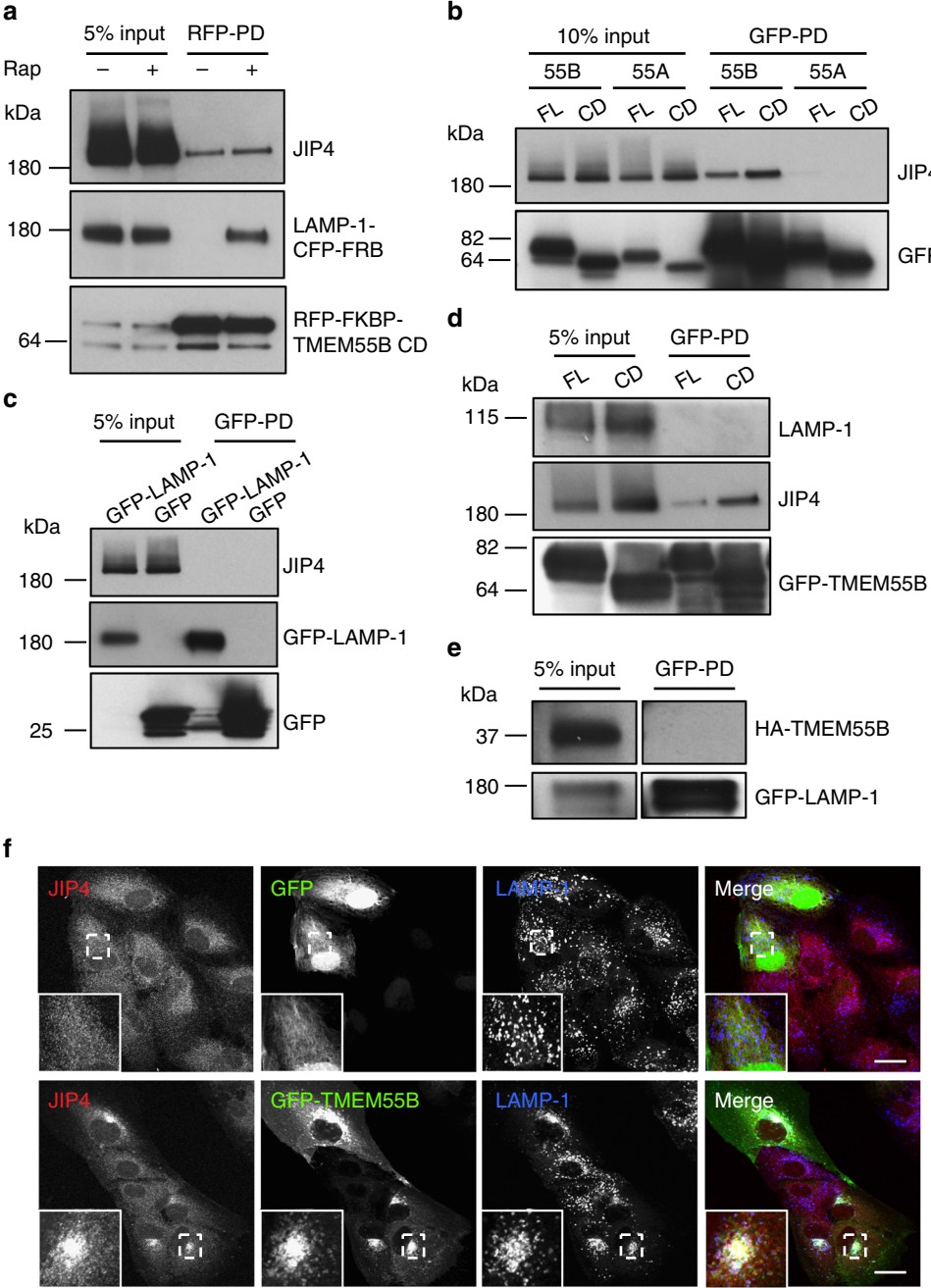

**Fig. 4** TMEM55B interacts with Dynein adapter JIP4. **a** Immunoblot of RFP pull-down from lysates of ARPE-19 cells co-transfected with LAMP-1-CFP-FRB and RFP-FKBP-TMEM55B CD and treated with 200 nM Rapamycin (Rap) (+) or vehicle (−) for 2 h. **b** Immunoblot of GFP pull-down from lysates of ARPE-19 cells transfected with GFP-TMEM55B FL (full length), GFP-TMEM55B CD (cytosolic domain), GFP-TMEM55A FL, or GFP-TMEM55A CD. **c** Immunoblot of GFP pull-down from lysates of ARPE-19 cells transfected with GFP-LAMP-1 or GFP. **d** Immunoblot of GFP pull-down from lysates of ARPE-19 cells transfected with GFP-TMEM55B FL or GFP-TMEM55B CD. **e** Immunoblot of GFP pull-down from lysates of ARPE-19 cells co-transfected with GFP-LAMP-1 and 3xHA-TMEM55B. **f** ARPE-19 cells were infected with adenovirus expressing GFP (top) or GFP-TMEM55B (bottom) for 24 h. Cells were fixed, permeabilized, and immunostained with antibodies against JIP4 and LAMP-1. Insets represent a 2.8-fold magnification of the indicated areas. Scale bars, 20 µm

Fig. 2b). This observation was supported by the retention of TMEM55B at the plasma membrane when endocytosis was inhibited by overexpression of a dominant-negative mutant of dynamin (GFP-Dynamin-K44A) (Supplementary Fig. 2c). Altogether, these results indicate that TMEM55B CD is required for targeting to lysosomes, and that a significant fraction of TMEM55B traffics through the plasma membrane before reaching the endo-lysosomal compartment.

Next, we used the FRB-FKBP rapamycin-induced heterodimerization system to localize TMEM55B CD to different intracellular compartments[40]. In this approach, TMEM55B CD was fused to the FKBP12 protein and, upon addition of rapamycin, the protein rapidly translocated to those organelles where its binding partner, the FRB domain, was targeted (Supplementary Fig. 3a). For lysosomal targeting, CFP tagged FRB was fused to LAMP-1 (Supplementary Fig. 3a). As expected,

RFP-FKBP-TMEM55B CD was rapidly recruited to lysosomes upon addition of rapamycin; and this was sufficient to induce lysosomes clustering in the perinuclear area (Fig. 3a).

To confirm these results, we fused FRB to the amino terminus of melanoregulin (Mreg; aa 1–42), which localizes to late endosomes/lysosomes via several acylated and palmitoylated residues[41] (Fig. 3b and Supplementary Fig. 3a). RFP-FKBP-TMEM55B CD was efficiently recruited to Mreg-positive vesicles following rapamycin treatment, causing dramatic clustering of late endosomes/lysosomes in the cell center (Fig. 3b). Transport

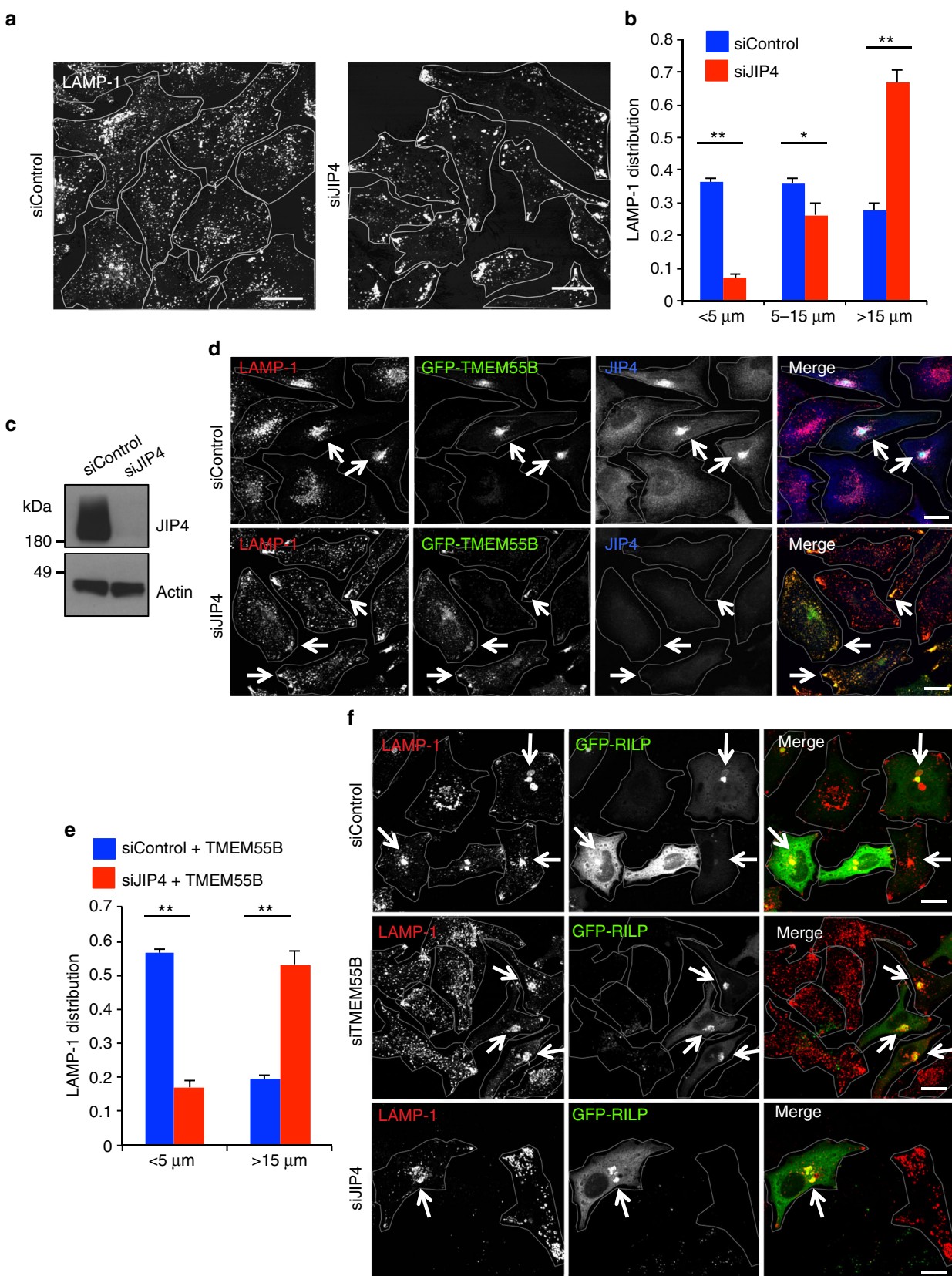

was exceedingly rapid, with retrograde movement detectable as early as 5 min following rapamycin induction (Supplementary Movie 1), and near complete clustering of endosomes after 30 min (Fig. 3b).

Finally, we tested the ability of TMEM55B CD to alter positioning of a non-secretory organelle. Mitochondria expressing the targeting sequence AKAP1-FRB-GFP experienced dramatic changes in mitochondria morphology after recruitment of RFP-FKBP-TMEM55B CD to outer membranes, changing from dispersed, inter-connected membranes (Fig. 3c, top panel) to a tight cluster in the cell center (Fig. 3c, bottom panel). We observed extensive co-localization of α-tubulin with mitochondria following RFP-FKBP-TMEM55B CD recruitment (Supplementary Fig. 3b), further suggesting that TMEM55B promotes microtubule minus-end organelle trafficking. Based on these results, we concluded that TMEM55B CD is both necessary and sufficient to drive retrograde trafficking of TMEM55B-targeted membranes, and does so by linking organelles to dynein motors.

**TMEM55B recruits JIP4 to lysosomal membranes**. Next, we sought to identify potential adapters that mediate interaction between TMEM55B and the dynein–dynactin complex. For this we co-expressed RFP-FKBP-TMEM55B CD and LAMP1-CFP-FRB and performed pull-down (PD) experiments in cells treated or non-treated with rapamycin, followed by mass spectrometry proteomic analysis. As a result, we identified the dynein scaffold JIP4 as a major interacting protein of TMEM55B CD (Supplementary Fig. 4a). To confirm these data, we expressed full-length GFP-TMEM55B and GFP-TMEM55B CD and tested for binding to JIP4. As expected, we detected binding of both proteins to endogenous JIP4 (Fig. 4b). TMEM55B exhibits a high degree of homology with TMEM55A, a protein that also localizes to late endosomes/lysosomes[36]. However, full length TMEM55A, as well as its N-terminal domain (TMEM55A CD), lacked the ability to bind JIP4 (Fig. 4b). Accordingly, expression of TMEM55A did not readily alter lysosome positioning (Supplementary Fig. 4b), further reinforcing the link between TMEM55B/JIP4 interaction and lysosomal clustering. As additional controls, we found that JIP4 did not interact with GFP or GFP-LAMP1 (Fig. 4c); likewise, TMEM55B did not bind endogenous or recombinant LAMP1 (Fig. 4d, e); confirming the specificity of the TMEM55B/JIP4 interaction.

It was previously described that JIP4 is primarily cytosolic; however, a fraction of the protein associates to endo/lysosomal membranes[13,42]. Interestingly, overexpression of GFP-TMEM55B resulted in a very noticeable accumulation of JIP4 on clustered lysosomes (Fig. 4f), suggesting that TMEM55B drives JIP4 membrane recruitment. Increased JIP4 and p150[Glued] recruitment to membranes upon TMEM55B overexpression was also observed by subcellular fractionation (Supplementary Fig. 4c, d). Altogether, our results suggest that TMEM55B recruits the motor adapter JIP4 to bring dynein–dynactin to lysosomal membranes.

**JIP4 depletion causes peripheral accumulation of lysosomes**. To further confirm the role of JIP4 in lysosomal dynamics, we transfected HeLa cells with siRNAs against JIP4 and analyzed lysosome distribution. JIP4-depleted cells exhibited almost complete peripheral accumulation of both lysosomes (Fig. 5a) and late endosomes (Supplementary Fig. 5c), with more than 60% of the LAMP-1 signal falling in the distal region of the cell (>15 μm from the nuclear rim) (Fig. 5b). Efficient JIP4 depletion was confirmed by western blot (Fig. 5c). Significant differences in lysosomal positioning in JIP4-depleted cells were confirmed by measuring LAMP-1 cumulative intensity distribution (Supplementary Fig. 5a). Moreover, expression of recombinant JIP4 in JIP4-depleted cells was sufficient to rescue the accumulation of lysosomes in the cell periphery (Supplementary Fig. 5b). Importantly, Golgi morphology and early endosome distribution was unaffected in cells depleted of either TMEM55B or JIP4 (Supplementary Fig. 5c, d), demonstrating the specificity of TMEM55B/JIP4 in regulating late-endosome/lysosome trafficking.

We next sought to determine if JIP4 is indeed the adaptor required to bring dynein–dynactin to TMEM55B-positive lysosomal membranes. As expected, siControl-treated cells expressing GFP-TMEM55B exhibited clustered lysosomes at the perinuclear area, as well as recruitment of cytosolic JIP4 to lysosome membranes (Fig. 5d, top). In contrast, expression of GFP-TMEM55B in cells depleted of JIP4 failed to induce clustering of lysosomes into the perinuclear region (Fig. 5d, bottom). Quantification of these experiments revealed that the percent of LAMP-1-positive structures in the distal region of the cell (>15 μm from the nuclear rim) changed from 20% in control cells to more than 50% in JIP4-depleted cells (Fig. 5e). Therefore, JIP4 is required to link TMEM55B-positive lysosomes to the dynein–dynactin complex.

Late-endosome/lysosome minus-end-directed trafficking has been extensively described via the Rab7/RILP/ORP1L pathway, where Rab7 effector RILP links Rab7-endosomes to dynactin p150[Glued], resulting in trafficking towards the MTOC[24]. In line with these observations, GFP-RILP expression in siControl-treated cells resulted in tight perinuclear clustering of LAMP-1 labeled lysosomes (Fig. 5f, top). Importantly, GFP-RILP was still capable of inducing perinuclear accumulation of lysosomes in cells depleted of either TMEM55B or JIP4 (Fig. 5f, middle and bottom), suggesting that RILP and TMEM55B function independently of one another. In agreement with this, GFP-TMEM55B clustered lysosomes in cells depleted of ORP1L or RILP (Supplementary Fig. 6a). Additionally, JIP4 displayed no interaction with either RILP or ORP1L (Supplementary Fig. 6b), and GFP-RILP expression was not capable of enriching recruitment of JIP4 to lysosome membranes, despite drastic lysosomal clustering (Supplementary Fig. 6c). MCOLN1 has also been implicated in lysosomal retrograde transport[28]. However, we found that TMEM55B induced efficient perinuclear accumulation of lysosomes in MCOLN1-depleted cells (Supplementary Fig. 7). Altogether, our results indicate that the Rab7/RILP/ORP1L

**Fig. 5** TMEM55B/JIP4 drive retrograde lysosomal trafficking independent of RILP. **a** HeLa cells treated with JIP4 or Control siRNA were immunostained with antibody against LAMP-1. **b** Quantification of lysosome distribution shown as the percentage of total fluorescence signal detected at 0–5 μm, 5–15 μm, or >15 μm from nuclear rim. Quantified results are presented as mean ± s.e.m. using two-tailed t-test *P < 0.05, **P < 0.005 were considered significant, n ≥ 30. **c** Immunoblot of HeLa cells depleted of JIP4 or Control RNAi. **d** HeLa cells treated with JIP4 or Control siRNA were infected with adenovirus expressing GFP-TMEM55B for 24 h. Cells were fixed, permeabilized, and immunostained with antibodies against LAMP-1. Arrows denote bulk of lysosomal accumulation. **e** Quantification of lysosome distribution, percentage of total fluorescence signal detected at 0–5 μm or >15 μm from nuclear rim. Quantified results are presented as mean ± s.e.m. using two-tailed t-test *P < 0.05, **P < 0.005 were considered significant, n ≥ 10. **f** HeLa cells depleted of TMEM55B, JIP4, or expressing control RNAis were transfected with GFP-RILP for 24 h. Cells were fixed, permeabilized, and immunostained with antibodies against LAMP-1. Arrows denote bulk of lysosomal accumulation. Scale bars, 20 μm

and MCOLN1–ALG2 pathways function independently of our newly described TMEM55B/JIP4 pathway for recruiting dynein to late endosomes/lysosomes to promote retrograde trafficking.

**Assessing TMEM55B phosphatidylinositol phosphatase activity.** A previous study suggested that TMEM55B functions as a type I phosphatidylinositol 4,5-bisphosphate 4-phosphatase, catalyzing the hydrolysis of 4-position phosphate on

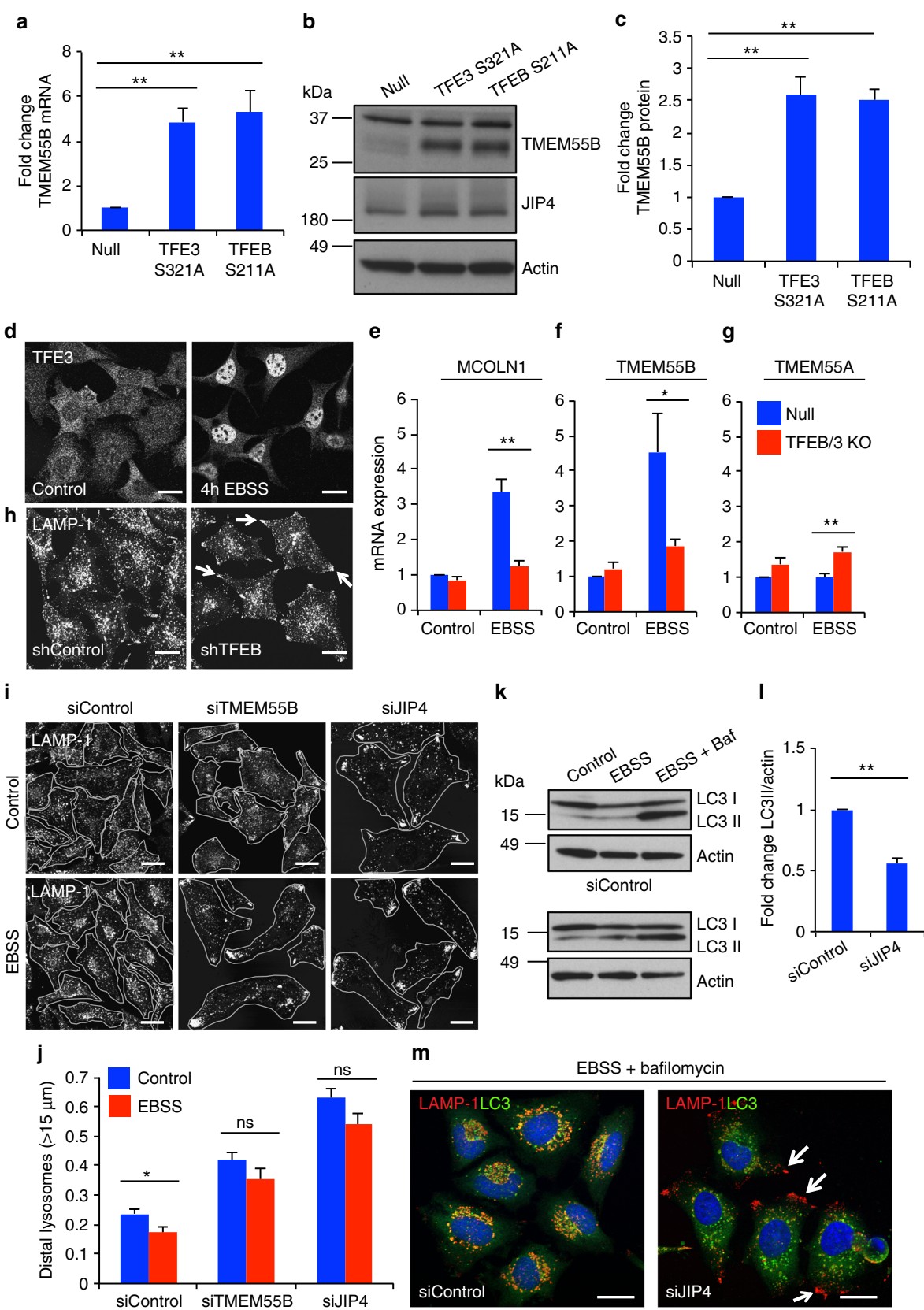

phosphatidylinositol-(4,5)-bisphosphate (PtdIns-4,5-P2) to generate phosphatidylinositol-5-phosphate (PtdIns-5-P or PI(5)P)[36]. Phosphatidylinositol phosphatases contain a $CX_5R$ motif, which is required for their catalytic activity. However, the $Cx_5R$ motif located in the CD domain of TMEM55B differs from other well-established phosphatases in that the arginine is followed by an isoleucine residue as opposed to a serine or threonine. Mutation of the $Cx_5R$ motif in TMEM55B full-length did not alter the ability of this protein to induce lysosomal clustering or interact with JIP4 (Supplementary Fig. 8a, b). Moreover, recruitment of TMEM55B CD to the plasma membrane via rapamycin-induced dimerization with $Lyn_{11}$-FRB-iRFP did not change PtdIns-4,5-P2 levels (Supplementary Fig. 8c, d). In contrast, recruitment of INPP5E to the cell surface, which was used as a positive control[43], induced a nearly complete loss of plasma membrane PtdIns-4,5-P2 (Supplementary Fig. 8c, d). Therefore, our results suggest that TMEM55B does not function as a bona fide PtdIns-4,5-P2 4-phosphatase and its role in retrograde trafficking is independent of its $Cx_5R$ motif.

To more rigorously test for phosphatase activity in TMEM55B and TMEM55A N-terminal domains, we purified GST-tagged proteins from *Escherichia coli* and assayed their activity against short-chain PtdIns-4,5-$P_2$. As a positive control, we used the lipid 5-phosphtatase SHIP2. We could not detect robust hydrolysis of PtdIns-4,5-$P_2$ by either TMEM55, though we could detect hydrolysis by SHIP2, despite the fact that this enzyme has by far the weakest activity against this substrate of the 5-phosphatases[44] (Supplementary Fig. 8e). Therefore, it seems that any lipid phosphatase activity of TMEM55B or TMEM55A is exceptionally weak, and unlikely to constitute the main function of these proteins.

**TFEB and TFE3 upregulate TMEM55B in response to starvation**. Previous studies have identified TMEM55B as a putative target of TFEB and TFE3[2,5,6]. To confirm this, HeLa cells were infected with adenovirus expressing constitutive active versions of TFEB (Ad-TFEB-S211A) and TFE3 (Ad-TFE3-S321A)[3,45]. As expected, TFEB and TFE3 caused a significant increase in TMEM55B mRNA and protein levels (Fig. 6a–c), thus confirming that these transcription factors modulate TMEM55B expression.

TFEB and TFE3 are activated by nutrient deprivation[3,4]. In fed cells, mTORC1-dependent phosphorylation promotes retention of TFEB and TFE3 in the cytosol. Following starvation, TFEB and TFE3 rapidly translocate to the nucleus where they induce transcription of multiple lysosomal and autophagic genes (Fig. 6d)[3,45–47]. Accordingly, incubation of MEFs in EBSS (starvation medium) caused significant MCOLN1 upregulation,

a lysosomal protein and TFEB/TFE3 target used in this experiment as a positive control (Fig. 6e). Starvation also significantly increased TMEM55B mRNA levels, and this increase was TFEB/TFE3-dependent since it was not observed in TFEB/TFE3 double knockout MEFs (Fig. 6f). A modest, although statistically significant, increase in JIP4 mRNA levels was also observed in nutrient-deprived cells (Supplementary Fig. 9a). In contrast, TMEM55A mRNA levels did not respond to starvation (Fig. 6g). This is in agreement with the fact that while four different CLEAR motifs have been identified in the TMEM55B promoter region (AACACGTGAC-288; GTCACGTGCA-193; GTCATGTGAC-154; ATCACGTGCT-36), analysis of the TMEM55A promoter did not reveal any obvious CLEAR motifs near to the transcription initiation site[2]. An increase in TMEM55B protein levels was also observed in HeLa cells incubated in EBSS for different periods of time (Supplementary Fig. 9b).

Lysosome transport to the perinuclear region following starvation is critical to facilitate autophagosome–lysosome interactions and controls the rate of autophagosome degradation. Interestingly, the degree of lysosomal clustering following starvation was reduced in TFEB-depleted HeLa cells, suggesting that TFEB regulates lysosomal positioning in response to nutrient levels (Fig. 6h and Supplementary Fig. 9c). Moreover, depletion of either TMEM55B or JIP4 was sufficient to block starvation-induced lysosomal clustering (Fig. 6i, j). Accordingly, autophagy flux was impaired in JIP4-depleted cells. In control cells, incubation with EBSS and bafilomycin led to a substantial accumulation of LC3 in perinuclear lysosomes (Fig. 6k–m). In contrast, $LC3_{II}$ levels following bafilomycin treatment were significantly reduced in JIP4-depleted cells (Fig. 6k, l) since autophagosomes were unable to fuse with peripheral lysosomes (Fig. 6m). Autophagy flux was also significantly impaired in TMEM55-depleted cells (Supplementary Fig. 9d). These results indicate that the TFEB/TMEM55B/JIP4 pathway is involved in mediating nutrient-dependent trafficking of lysosomes to stimulate efficient autophagic flux.

**TMEM55B responds to changes in cholesterol levels**. A previous study suggested that the levels of TMEM55B are modulated by the transcription factor SREBF2 in response to cholesterol levels[48]. In agreement with this report, we found that in HeLa cells, sterol depletion by treatment with a combination of lipoprotein depleted serum and statins caused a significant TMEM55B upregulation both to the mRNA and protein levels (Fig. 7a, b). Upregulation of TMEM55B was also observed following treatment with U18666A (Fig. 7a, b), an amphipathic

**Fig. 6** TFE3 and TFEB increase TMEM55B expression to cluster lysosomes during starvation. **a–c** ARPE-19 cells were infected with adenovirus expressing TFE3-S321A-Myc, TFEB-S211A-FLAG, or null virus for 36 h. **a** Relative quantitative real-time PCR analysis of TMEM55B mRNA transcript levels (mean ± s.e. m. of the RNA fold change of indicated TMEM55B normalized to GAPDH mRNA, $n = 4$). **b** Representative immunoblot of lysates from TFE3-S321A-Myc and TFEB-S211A-FLAG expressing cells. **c** Quantification of TMEM55B protein levels. **d** Control MEFs untreated or starved in EBSS for 4 h. Cells were fixed, permeabilized, and immunostained with antibodies against TFE3. **e–g** Relative quantitative real-time PCR analysis of **e** MCOLN1, **f** TMEM55B and **g** TMEM55A mRNA transcript levels from null- or TFE3/TFEB knockout MEFs, untreated or starved in EBSS for 4 h (mean ± s.e.m. of the RNA fold change of indicated TMEM55B normalized to GAPDH) $n = 6$ from three independent experiments. **h** HeLa cells stably expressing control or TFEB shRNAs were starved in EBSS for 4 h. Cells were fixed, permeabilized, and immunostained with antibodies against LAMP-1. **i** HeLa cells expressing control, TMEM55B, and JIP4 RNAis were starved in EBSS for 4 h. Cells were fixed, permeabilized, and immunostained with antibodies against LAMP-1. **j** Quantification of lysosome distribution, percentage of total fluorescence signal detected >15 μm from nuclear rim. Quantified results are presented as mean ± s.e.m. using two-tailed $t$-test *$P < 0.05$, **$P < 0.005$ were considered significant, $n \geq 30$. **k** Immunoblot of lysates from HeLa cells treated with siControl or siJIP4 siRNA, then starved in EBSS for 4 h, or starved in EBSS for 4 h with 2 h 100 nM bafilomycin. Note that all the samples were run in the same gel but are presented in two panels due to space limitations. **l** quantification of $LC3_{II}$/Actin ratios from siControl or siJIP4-treated HeLa cells starved in EBSS 4 h with 100 nM bafilomycin for 2 h. Quantified results are fold increase of $LC3_{II}$/Actin from siControl after starvation and bafilomycin treatment and data are presented as mean ± s.e.m. using two-tailed $t$-test **$P < 0.005$, $n = 3$. **m** HeLa cells treated with siControl or siJIP4 siRNA and starved with EBSS for 4 h and 100 nM bafilomycin for 2 h. Cells were fixed, permeabilized, and immunostained with antibodies against LAMP-1, LC3, and DAPI. Scale bars, 20 μm

steroid which blocks cholesterol exit from late endosomes/lysosomes, inducing lysosomal stress[49]. U18666A also inhibits cholesterol biosynthesis by inhibiting oxidosqualene cyclase and desmosterol reductase[50]. Interestingly, U18666A induced TFE3 activation and TFEB/TFE3-dependent TMEM55B upregulation in MEFs, suggesting that the TFEB/TMEM55B/JIP4 pathway may also participate in the adaptation of lysosomal function and distribution in response to cholesterol (Fig. 7c, d). Accordingly, depletion of TMEM55B, or more dramatically JIP4, prevented U18666A-mediated lysosomal clustering without affecting

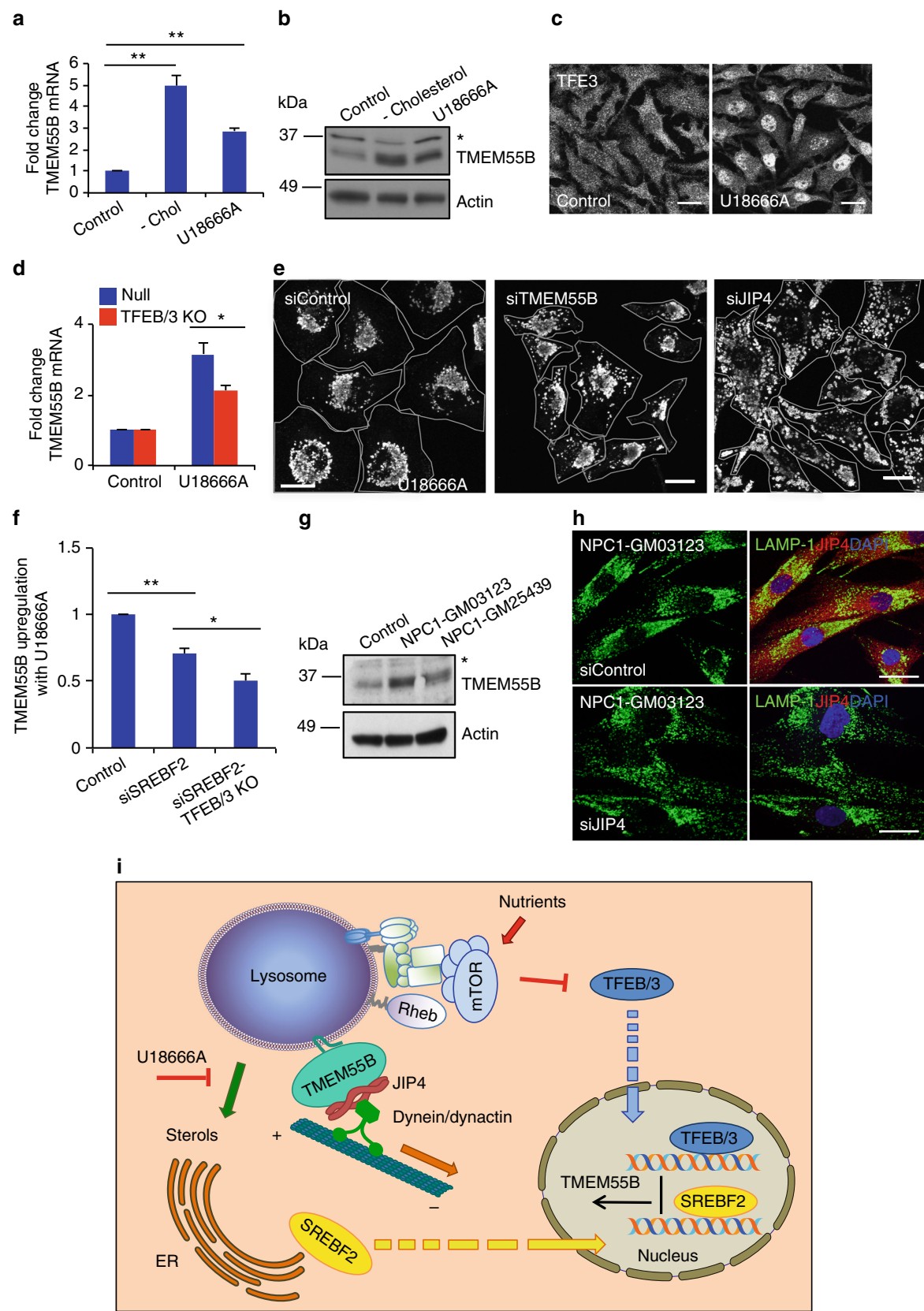

U18666A-induced lysosomal enlargement (Fig. 7e and Supplementary Fig. 9e). In agreement with the report by Medina et al.[48], we observed that the upregulation of TMEM55B induced by U18666A was significantly decreased in cells depleted of SREBF2 (Fig. 7f); however, the decrease was even more pronounced in cells depleted of SREBF2, TFEB, and TFE3 (Fig. 7f), suggesting that the three transcription factors contribute to regulate TMEM55B levels in response to changes in cellular cholesterol homeostasis.

It is well established that U18666A mimics the lack of functional Niemann-Pick type C protein responsible for the NPC disease. Therefore, we hypothesized that the expression of TMEM55B might be altered in NPC cells. Analysis of control (clone GM25445) and NPC-1 fibroblasts (clones GM25439 and GM03123) confirmed that TMEM55B protein levels were upregulated in patient cells (Fig. 7g). Similar to our observations in U18666A-treated cells, JIP4 depletion reduced perinuclear clustering of lysosomes in NPC fibroblasts (Fig. 7h, Supplementary Fig. 9f, g), suggesting that alterations in the TMEM55B/JIP4 pathway may contribute to the pathology of the disease.

In summary, we conclude that the regulation of TMEM55B expression in response to changes in nutrient and cholesterol levels is critical to ensure proper cellular homeostasis (Fig. 7i).

**MAPK-dependent regulation of the TMEM55B/JIP4 pathway**. We noticed that treatment of cells with different compounds known to generate oxidative stress and lysosomal damage resulted in a rapid and pronounced redistribution of lysosomes toward the cell center (Fig. 8). This suggests that besides the transcriptional upregulation of TMEM55B, additional post-translational mechanisms may contribute to the regulation of the TMEM55B/JIP4/dynein–dynactin pathway. Treatment of MEFs with curcumin, a pro-apoptotic agent that induces mitochondrial-dependent production of reactive oxygen species (ROS) and lysosomal destabilization[51–54], resulted in a very obvious clustering of lysosomes as early as two hours following the addition of the drug (Fig. 8a). Increased lysosomal retrograde transport was also observed when MEFs were treated with other ROS-generating compounds such as sodium arsenite ($NaAsO_2$)[55] (Supplementary Fig. 10a). Curcumin induced robust recruitment of JIP4 and the dynactin subunit p150[Glued] to lysosomal membranes (Fig. 8b, c), suggesting that TMEM55B/JIP4 complex may regulate lysosomal retrograde transport under acute stress conditions. Accordingly, curcumin and $NaAsO_2$-induced lysosomal clustering was inhibited in JIP4-depleted MEFs (Fig. 8d and Supplementary Fig. 10a). JIP4-dependent lysosomal retrograde transport in response to curcumin and $NaAsO_2$ was confirmed in U2OS cells (Supplemental Fig. 10b). Treatment of U2OS cells with $NaAsO_2$ resulted

in a clear shift in TMEM55B electrophoretic mobility, as well as an increase in TMEM55B dimerization (Fig. 8e), suggesting that TMEM55B might undergo post-translational modifications in response to acute stress.

Previous studies reported that JIP4 functions as a scaffold protein, playing a crucial role in the regulation of mitogen-activated protein kinase (MAPK) signaling cascades. In particular, JIP4 binds, promotes the activation, and is phosphorylated by p38 MAPK[56,57]. This opened up the question of whether or not lysosomal transport is regulated by MAPKs. We found that curcumin caused a rapid activation of different MAPKs, including p38, JNK, and Erk1/2 (Fig. 8f); therefore, we assessed whether any of these kinases might contribute to changes in lysosomal distribution. Interestingly, lysosomal clustering and JIP4 recruitment to lysosomes was prevented by incubation with p38 inhibitors, but not JNK or Erk1/2 inhibitors (Fig. 8g, h), thus revealing a novel and unexpected role of p38 in retrograde organelle transport. Altogether, our data suggest that the TMEM55B/JIP4 complex links signaling to rapid changes in lysosomal positioning in response to cellular damage.

## Discussion

Lysosome positioning is increasingly acknowledged as a critical determinant of many physiological processes including plasma membrane repair, antigen presentation, adaptation to nutrient availability, cholesterol homeostasis, and cell migration. Moreover, alterations in lysosomal distribution have been associated with a variety of pathological conditions, including cancer, neurological disorders, and lysosomal storage diseases.

Here we characterize a novel mechanism of lysosomal positioning regulation. We found that the lysosomal protein TMEM55B recruits JIP4 to lysosomal membranes to promote dynein-dependent retrograde transport (Fig. 7i). Therefore, TMEM55B functions as a linker between lysosomes and microtubules. JIP4 belongs to a family of motor adapter proteins, together with JIP3, which possess the unique ability to bind both kinesin-1 and dynactin[42,58–60]. JIP4 has been implicated in the delivery of recycling endosomes to the midbody during cytokinesis, as well as in the transport of membrane-tethered membrane type 1-matrix metalloproteinase (MT1-MMP)-positive endosomes to invadopodia[13,42]. Meanwhile, JIP3, which has a preferential neuronal expression, has been implicated in axonal retrograde transport of lysosomes in zebrafish[61]. In these instances, the switching between dynein–dynactin and kinesin-1 is regulated by the small GTPase Arf6, which interacts with the second leucine domain of JIP4/3 in its GTP-bound active state, interfering the binding of JIP4/3 to kinesin-1[62].

**Fig. 7** TFEB/3 and SREBF2 co-operate to regulate TMEM55B levels in response to changes in cholesterol levels. **a, b** HeLa cells were treated with drugs to deplete cellular cholesterol for 120 h, treated with 10 μM U18666A for 18 h, or left untreated. **a** Relative quantitative real-time PCR analysis of TMEM55B mRNA transcript levels (mean ± s.e.m. of the RNA fold change of indicated TMEM55B normalized to GAPDH mRNA, using two-tailed t-test **$P < 0.005$ were considered significant) $n = 3$. **b** Representative immunoblot of lysates from cholesterol depleted cells. **c** Control MEFs were untreated or incubated in 10 μM U18666A for 18 h. Cells were fixed, permeabilized, and immunostained with antibodies against TFE3. **d** Relative quantitative real-time PCR analysis of TMEM55B mRNA transcript levels from null- or TFE3/TFEB knockout MEFs, untreated or starved in EBSS for 4 h (mean ± s.e.m. of the RNA fold change of indicated TMEM55B normalized to GAPDH, using two-tailed t-test *$P < 0.05$ were considered significant) $n = 6$ from three independent experiments. **e** HeLa cells depleted of TMEM55B, JIP4, or control RNAi were starved in EBSS for 4 h and then immunostained with antibody against LAMP-1. **f** Control MEFs treated with control and SREBF2 siRNAs and TFEB/TFE3 KO MEFs treated with SREBF2 siRNAs were incubated with 10 μM U18666A for 18 h and the TMEM55B mRNA transcript levels were quantified by qRT-PCR (mean ± s.e.m. of the RNA fold change normalized to TMEM55B levels in siControl-treated cells, using two-tailed t-test *$P < 0.05$, **$P < 0.005$ were considered significant) $n = 6$ from three independent experiments. **g** Immunoblot of lysates from primary skin fibroblasts from Niemann-Pick C (NPC1) patient 1, NPC1 patient 2, and genetic matched control patient. **h** NPC1 patient fibroblasts depleted of JIP4 with RNAi. Cells were fixed, permeabilized, and immunostained with antibodies against LAMP-1, JIP4, and DAPI. **i** Model of TMEM55B transcriptional regulation. Nutrient deprivation inactivates mTOR to allow TFEB/3 translocation to the nucleus to activate transcription of TMEM55B and induce lysosome

Lysosomes undergo bidirectional transport along microtubules through the opposite activities of dynein and kinesins. Here we postulate that the interaction of TMEM55B with the JIP4–dynactin complex helps bring dynein to the lysosomal surface, thus promoting retrograde transport. Accordingly, the balance between dynein and kinesin is perturbed by TMEM55B silencing or overexpression, resulting in transport of lysosomes toward the periphery or the cell center, respectively. It remains to

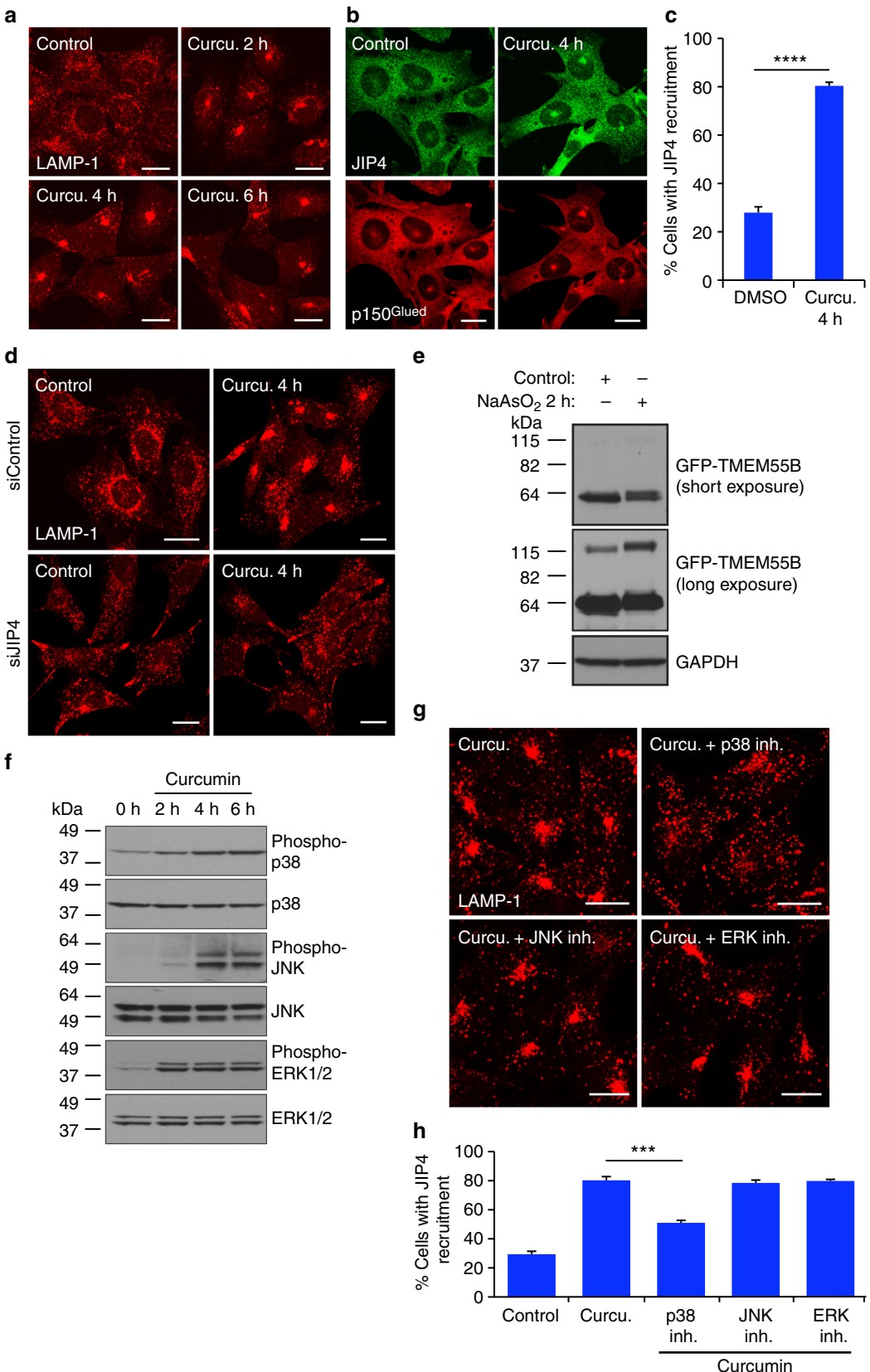

be determined whether TMEM55B functions as a mere JIP4 recruiter or if it may also stimulate the interaction between JIP4 and the dynein–dynactin complex.

Lysosomes play a critical role in maintaining cellular homeostasis and helping cells adapting to different stressors. By regulating TFEB and TFE3 activation, cells modulate both the number and the activity of lysosomes. We found that TFEB and TFE3 also control TMEM55B protein levels, indicating that these transcription factors play an active role in lysosomal positioning regulation. This regulation seems to be particularly important during starvation. Following nutrient deprivation, TFEB and TFE3 rapidly translocate from the cytosol to the nucleus, where they upregulate expression of multiple lysosomal genes, including TMEM55B. Increased TMEM55B levels will favor movement of lysosomes toward the center of the cell, promoting fusion between lysosomes and autophagosomes to ensure efficient autophagic flux. In agreement with our results, a recent study showed that TFEB overexpression increases the perinuclear distribution of LAMP-1 vesicles[63]. Perhaps not surprisingly, cells have developed additional mechanism to guarantee autophagosome degradation. MCOLN1, also a TFEB/TFE3 target, mediates retrograde lysosomal transport through interactions with the ALG-2/dynein–dynactin complex[28]; and recruitment of folliculin to lysosomes under starvation facilitates the binding between RILP and Rab34, promoting retention of lysosomes in the cell center[38]. It will be important to determine how these independent mechanisms work in conjunction to ultimately regulate lysosomal trafficking.

TFEB/3, TMEM55B, and JIP4 also participate in the cholesterol-dependent distribution of lysosomes. LDL-bound cholesterol enters cells via endocytosis and it is delivered to lysosomes to be converted into unesterified cholesterol. Free cholesterol is then transferred from lysosomes to other cellular compartments in a process that requires transport and tethering of late endosomes/lysosomes with the endoplasmic reticulum (ER) membranes. Inhibition of cholesterol efflux from lysosomes in U18666A-treated cells was sensed by different cellular compartments, promoting activation of SREBF2 at the ER and TFEB/3 at the lysosomes. The resulting increase in TMEM55B levels promoted transport of lysosomes to the perinuclear region and likely facilitated lysosome/ER interactions. In NPC1 and other lysosomal diseases, the persistent accumulation of cholesterol in late endosomes/lysosomes leads to aberrant lysosomal clustering in the cell center. Interestingly, JIP4 silencing reduced clustering in NPC1 cells, suggesting that this pathway may be involved in the accumulation of the cholesterol-laden lysosomes observed in lysosomal diseases. The regulation of cholesterol egress from late endosomes/lysosomes likely requires the coordinated action of multiple cholesterol sensing and transporting molecules, as is the case of the Rab7/RILP/ORP1L pathway. One possibility is that the TMEM55B/JIP4 and Rab7/RILP/ORP1L localize to a subset of late endosomes/lysosomes. Alternatively, both complexes may exhibit different roles in cholesterol sensing, trafficking and transfer.

Recruitment of JIP4 to lysosomes might have additional roles. JIP4 is a scaffold protein that interacts with c-Jun NH$_2$-terminal kinase (JNK) and functions as an activator of the p38 mitogen-activated protein (MAP) kinase pathway[56], suggesting that JIP4 might link transport of lysosomes to signaling. In agreement with this idea, we found that induction of acute stress by treatment with NaAsO$_2$ or curcumin resulted in post-transcriptional regulation of TMEM55B, increased JIP4 and p150$^{Glued}$ recruitment to lysosomes, and rapid lysosomal retrograde transport via a mechanism that required JIP4 and p38 activation. These results show interesting similarities with previous studies indicating that Sunday Driver (syd), the JIP3/JIP4 orthologue in *Drosophila*, links retrograde vesicular transport to JNK signaling in axons following injury[59]. In this case, activation of JNK following nerve injury leads to syd phosphorylation and increased syd/dynactin association, resulting in a switch to retrograde transport[59]. Our data suggest that the TMEM55B/JIP4 complex may be part of a surveillance system which allows rapid changes in lysosomal positioning in response to cellular damage. Moreover, TMEM55B regulation, both to the transcriptional and post-translational levels, may help coordinating the lysosomal response to different types of stress.

In summary, our work adds a new component to the repertoire of protein complexes implicated in the dynamic regulation of lysosomal positioning. The emerging number of proteins implicated in this process suggest the possibility of some degree of specialization, with some complexes playing a primary role in certain cell types (e.g., neuronal vs. not neuronal) or in response to specific stimuli. In this regard, the TMEM55B/JIP4/dynein–dynactin pathway is particularly interesting, since it exhibits a dual function. It regulates the constitutive bi-directional transport of lysosomes but it also acts "on-demand" to promote retrograde lysosomal transport under certain stress conditions. Given the emerging implications of lysosomal positioning in different human diseases, we propose that the TMEM55B/JIP4 pathway may represent a promising target for future drug intervention.

## Methods

**Cell lines and treatments**. ARPE-19 cells (American Type Culture Collection, ATCC) were grown at 37 °C, 5% CO$_2$ in a 1:1 mixture of Dulbecco's Modified Eagle Medium (DMEM) and Ham's F12 media with Glutamax supplemented with 10% fetal bovine serum (FBS) (Thermo-Fisher) and 100 U/ml penicillin and 100 μg/ml streptomycin (Gibco). HeLa cells (ATCC) and U2OS cells (ATCC) were grown at 37 °C, 5% CO$_2$ in DMEM with glutamax supplemented with 10% fetal bovine serum (Thermo-Fisher) and 100 U/ml penicillin and 100 μg/ml strepto-mycin (Gibco). Control and TFE3/B KO MEFs were generated by transducing wild-type MEF cells with lentiviruses containing control or TFEB and TFE3 CRISPR–Cas9 guide RNA-targeting sequences[6]. Stable shMCOLN1 HeLa cells were generated by infection with MCOLN1 lentiviral shRNA[64]. HeLa shTFEB were generated by infection with TFEB lentiviral shRNA[3]. Skin fibroblasts from Niemann Pick-C (NPC) patients (NPC1-1: GM25439, NPC1-2: GM03123) and

**Fig. 8** JIP4-dependent retrograde lysosomal transport in response to acute oxidative stress. **a**, **b** MEFs were incubated with 20 μM curcumin for the indicated times. Cells were fixed, permeabilized, and immunostained with antibodies against LAMP-1 (**a**) or JIP4 and p150$^{Glued}$ (**b**). **c** Quantification of cells with JIP4 recruitment to lysosomes in DMSO or curcumin treated MEFs. DMSO $n = 745$, curcumin $n = 805$ from three independent experiments. Error bars denote s.e.m. *P* value calculated using two-tailed *t*-test ****$P < 0.0001$. **d** MEFs treated with control or JIP4 siRNAs and incubated with curcumin for 4 h. Cells were fixed, permeabilized, and immunostained with antibody against LAMP-1. **e** Immunoblot of U2OS expressing GFP-TMEM55B cells untreated or treated with 150 μM NaAsO$_2$. **f** Immunoblot of lysates from MEFs treated with curcumin for the indicated times. **g** MEFs were pre-incubated with MAPK inhibitors, followed by combination of inhibitors and curcumin. Cells were fixed, permeabilized, and immunostained with antibody against LAMP-1. **h** Quantification of cells with JIP4 recruitment to lysosomes in curcumin/MAPK inhibitor-treated MEFs. Control $n = 568$, curcumin $n = 583$, curcumin + p38 inhibitor (20 μM) $n = 598$, curcumin + JNK inhibitor (25 μM) $n = 590$, curcumin + ERK inhibitor (20 μM) $n = 575$ from two independent experiments. Error bars denote s.e.m. *P* value calculated using one-way ANOVA ***$P < 0.001$. Scale bars, 20 μm

matching control (GM25445) were obtained from the Coriell Institute for Medical Research (New Jersey, USA) and were maintained in Dulbecco's modified Eagle's medium with glutamax (DMEM) (Thermo-Fisher) supplemented with 100 U/ml penicillin (Gibco) and 15% (w/v) fetal bovine serum (FBS). COS-7 cells (CRL-1651) were obtained from ATCC and were maintained in Dulbecco's modified Eagle's medium with glutamax (DMEM) (Thermo-Fisher) supplemented with 100 U/ml penicillin (Gibco) and 10% (w/v) fetal bovine serum (FBS).

For starvation experiments, cells were washed three times in Hank's balanced salt solution (Thermo-Fisher) and incubated for 2–6 h at 37 °C in Earle's balanced salt solution (Starvation media) (Thermo-Fisher). For rapamycin-inducible dimerization experiments, cells were incubated in 200 nM rapamycin (Cell Signaling Technologies) for the indicated times. For microtubule depolymerizing assays, cells were incubated in 10 μM nocodazole for 2 h at 37 °C, then shifted to ice for 30 min in the presence of nocodazole. For nocodazole washouts, cells were washed and incubated in nocodazole-free culture medium for the indicated times. For cholesterol depletion, cells were incubated in DMEM with glutamax supplemented with 10% lipo-protein deficient serum (Khaled biological), 50 μM lovastatin (Sigma) and 230 μM mevalonate (Sigma) for 120 h. For induction of cholesterol accumulation in lysosomes, cells were treated with 10 μM U18666A for 18 h. For acute oxidative stress in MEFs, cells were treated with 20 μM curcumin (Sigma) for 2–6 h, or 150 μM sodium-(meta)arsenite;NaAsO$_2$ (Santa Cruz Biotechnology) for 2 h. For acute oxidative stress in U2OS, cells were treated with 40 μM curcumin for 6 h, or 300 μM NaAsO$_2$ for 2 h. For MAPK inhibition, cells were pretreated with inhibitors for 30 min at indicated concentrations, then treated with a combination of inhibitor and either curcumin or NaAsO$_2$ as indicated above. MAPK inhibitors: ERK inhibitor: U0126 (Selleckchem) 20 μM, JNK inhibitor: SP600125 (Selleckchem) 25 μM, p38 inhibitor: SB203580 (Selleckchem) 20 μM.

**Recombinant DNA constructs.** Mammalian expression constructs were generated using standard molecular biology techniques or obtained as generous gifts. GFP-TMEM55A and GFP-TMEM55B expression vectors were generated by cloning the full-length encoding sequence of human TMEM55A and TMEM55B obtained by PCR amplification from human heart marathon-ready cDNA (Takara Bio USA, Inc.) followed by in-frame cloning into EcoRI-SalI sites of pmEGFP-C2 (Takara Bio USA, Inc.) or pCiNeo-3xHA with a GFP or 3 in tandem HA tags fused to the amino-termini of TMEM55B respectively. The TMEM55B CD construct was generated by the insertion of a stop codon after the tyrosine 207 codon of GFP-TMEM55B expression vector. The TMEM55B TM construct was generated by cloning the encoding sequence of TMEM55B (cysteine 197 to serine 277) obtained by PCR amplification, followed by in-frame cloning into EcoRI-SalI sites of pmEGFP-C2. TMEM55A CD was generated by PCR amplification of pmEGFP-TMEM55A with primers designed to remove residues 182–257 with EcoRV restriction sites. pmRFP-FKBP-TMEM55B-CD expression vector was generated by cloning TMEM55B-CD into PvuI-BamHI sites of pmRFP-FKBP (provided by Dr. Tamas Balla, NIH, Bethesda, USA). pMreg-FRB-HA-CFP and pLAMP1-FRB-HA-CFP expression vectors were generated by cloning the encoding sequence of the first 42 amino acids of mouse melanoregulin (provided by Dr. John Hammer, NIH, Bethesda, USA) or the full-length encoding sequence of human LAMP1 obtained by PCR amplification respectively, followed by in-frame cloning into NheI-BglII sites of pPM-FRB-HA-CFP (provided by Dr. Tamas Balla, NIH, Bethesda, USA). Mitochondrial A-kinase anchor protein 1 (AKAP1)-FRB-HA-GFP was generated by PCR amplification of AKAP1-FRB-HA-RFP expression vector (provided by Dr. Tamas Balla, NIH, Bethesda, USA) with primers designed to remove RFP, followed by in-frame cloning of GFP cDNA into EcoRV and KpnI restriction sites. Amino acid substitutions in TMEM55B were made using the QuickChange Lightning site-directed mutagenesis kit (Stratagene) according to the manufacturer's instructions. GFP-P50 Dynamitin (provided by Dr. Tina Schroer, Johns Hopkins University), GFP-Dynamin K44A (provided by Dr. Juan Bonifacino, NIH), pmRFP-FKBP-INPP5E (provided by Dr. Tamas Balla, NIH) and GFP-RILP and GFP-ORP1L (Dr. Jacques Neefjes, Leiden University, The Netherlands) were generously provided by the indicated investigators. pEGFP-N1-PH-PLCD1 (PMC2132833), piRFP-Lyn$_{11}$-FRB[65], and pmCherry-C1-FKBP-INPP5E[43] was as previously described, except for the latter wherein mCherry replaced mRFP. pCMV6-JIP4(SPAG9)-Myc-Flag (OriGene). pGEX-5X1-TMEM55B-CD expression vector was generated by cloning TMEM55B-CD into EcoRI-SalI sites of pGEX-5X1 (GE Healthcare Life Sciences).

Plasmids were isolated from bacterial using the NucleoBond Xtra Midi Plus kit (Machery-Nagel) or QIAprep Spin Miniprep Kit (Quiagen).

**Cell transfections.** Plasmid transfections were performed using Lipofectamine 2000, Lipofectamine 3000 (Thermo-Fisher), or Fugene 6 (Roche), according to manufacturer's protocol.

**Adenovirus.** Adenovirus expressing GFP-TMEM55B, TFEB-S211A-Flag, and TFE3-S321A-Myc were prepared, amplified, and purified by Welgen, Inc.[3]

**RNA interference.** Cells were transfected using Lipofectamine RNAimax (Thermo-Fisher) transfection reagent and 100 nM of ON-TARGETplus non-targeting pool siRNA duplexes, ON-TARGETplus smart pool siRNA duplexes

targeted against human RILP or mouse SREBF2 and JIP4 genes (Dharmacon-Thermo Scientific) or custom sequences were used for the following genes: human TMEM55B (GCAUCAGCAUGUAGUAGUCAAAUUUU), human TMEM55A (GGACACAUCUCGGCGAAUA), human ORP1L (UGCCAGUGCCGGAUU-CUGAUUUU), and human JIP4 (#1 GAGCAUGUCUUUACAGAUCUU and #4 GCAUCACAGUGGUUGGUUGUU). Seventy two hours after second round of transfection cells were kept in complete medium (Control), incubated in Earle's balanced salt solution (4 h EBSS), or treated with 10 μM U18666A for 18 h before being harvested for analysis. Experiments involving siRNA-induced knockdown of indicated proteins are representative of three or more independent experiments.

**Lysosome distribution quantification.** Linear profile distribution: Lysosome distribution was quantified using Fiji plot profile. LAMP-1 signal from individual cells was quantified by drawing a line (width=10 pixels/3 μm) from the cells nuclear rim to the cell border using a cytosolic protein marker. Averaged percentage distribution at 0–5 μm from nuclear rim (perinuclear), 5–15 μm from nuclear rim (intermediate), and total signal 15 μm to cell border (peripheral) was calculated per cell by dividing the fractioned signal intensity from the total line fluorescence. Three linear profiles extending from the rim of the nucleus were averaged per cell. Line scans were drawn in various regions of the cell to accurately sample total LAMP-1 distribution. No portion of the cell was measured more than once. Thirty or more cells were quantified per cell treatment and averaged to quantify the population lysosome distribution.

Radial profile distribution: Lysosome distribution was quantified using Fiji radial profile. LAMP-1 signal from individual cells was quantified by drawing a circle around the entire volume of the cell. Individual cell profiles were compared side by side.

Cumulative intensity distribution: Lysosome distribution was quantified using Fiji. Individual cell ROI's were outlined using a cytosolic protein marker and whole cell LAMP-1 signal fluorescence was measured. ROI was then decreased by 10% and LAMP-1 signal fluorescence was measured at each decrease. To generate a LAMP-1 distribution curve, the signal intensity of each fraction was divided by the total cell signal. Twenty or more cells were quantified per cell treatment and averaged to quantify the population lysosome distribution.

**Immunofluorescence staining.** Cells grown on glass coverslips were washed with PBS and fixed in 4% paraformaldehyde at room temperature for 15 min or 1:1 methanol/acetone at −20 °C for 15 min. After fixation, cells were washed with PBS and then incubated with the indicated primary antibodies in IF buffer (PBS containing 10% FBS and 0.1% (w/v) saponin) for 1 h at room temperature. Cells were washed three times with PBS and incubated with the corresponding secondary antibodies in IF buffer for 30 min at room temperature. PBS washed coverslips were mounted onto glass slides with Prolong Diamond antifade with Dapi (Life Technologies). List of antibodies and their dilutions in Supplementary Table 1. Cell images are representative from three or more independent experiments.

**Time-lapse imaging.** Time-lapse confocal microscopy of ARPE-19 cells transiently co-expressing pmRFP-FKBP-TMEM55B CD and pMreg-FRB-HA-CFP was performed in medium containing either DMSO or rapamycin (0.5 μM). Confocal images of RFP-FKBP-TMEM55B CD were acquired every 7 s for a duration of 18 min on a Zeiss LSM 510 confocal system with a ×63 NA 1.4 oil immersion objective using the 543 nm laser excitation (Carl Zeiss) and equipped with a live cell imaging chamber. Images were processed using NIH ImageJ software.

**Electrophoresis and immunoblotting.** Cells were washed with ice-cold PBS, resuspended in lysis buffer (25 mM Tris-HCl, pH 7.5, 125 mM NaCl, 1% Triton X-100 (wt/vol)) supplemented with protease and phosphatase inhibitors cocktail (Roche), and lysed by passing the samples ten times through a 25 G needle. Cell lysates were centrifuged at 16,000 × g for 10 min at 4 °C, and the soluble fractions were collected. Samples were analyzed by SDS-PAGE (4–20% gradient gels, Thermo-Fisher) under reducing conditions and transferred to nitrocellulose. Membranes were immunoblotted using the indicated antibodies. Horseradish peroxidase-chemiluminiscence was developed by using Western Lightning Chemiluminescence Reagent Plus (PerkinElmer Life Sciences). For a list of antibodies and their dilutions see Supplementary Table 1. Western blots are representative of three or more independent experiments.

**RNA isolation and relative quantitative real-time PCR.** RNA was isolated by using PureLink RNA Mini Kit (Thermo-Fisher) following manufacturer's recommendations. One to two micrograms of RNA were reverse transcribed in a 20 μl reaction using oligo(dT)[20] and SuperScript III First-Strand Synthesis System (Thermo-Fisher). Relative quantitative real-time PCR was performed in a total reaction volume of 10 μl, using 2 μl (20 μg/μl) cDNA, 1 μl gene specific primer mix (QuantiTect primer Assays), 5 μl SYBR GreenER qPCR SuperMix (Thermo-Fisher), and 2 μl water. The quantification of gene expression was performed using 7900HT Fast Real-Time PCR System (Applied Biosystems) in triplicate. The thermal profile of the reaction was: 50 °C for 2 min, 95 °C for 10 min and 45 cycles of 95 °C for 15 s followed by 60 °C for 1 min. Amplification of the sequence of interest was normalized with a reference endogenous gene Glyceraldehyde 3-

phosphate dehydrogenase (GAPDH). The value was expressed as a fold change relative to RNA from cells infected with control adenovirus (Ad. Null), control siRNA (siControl), or non-treated cells. For data analysis, the 7900HT Fast Real-Time PCR System Software was used (Applied Biosystems).

**GFP and RFP pull-down experiments**. Cells were washed with ice-cold PBS, resuspended in buffer containing 25 mM Tris-HCl, pH 7.5, 300 mM NaCl, 5 mM EDTA, and 1% Triton X-100 (wt/vol) supplemented with protease inhibitor cocktail, and lysed by passing the samples 10 times through a 25 G needle. Cell lysates were centrifuged at $16,000 \times g$ for 10 min at 4 °C and soluble fractions (S16) were isolated. For GFP or RFP pull-downs, S16 was incubated with GFP- or RFP-nanobody beads (Chromotek) for 2 h at 4 °C with constant rotation. Unbound material was removed and beads were washed with buffer containing 25 mM Tris-HCl, pH 7.5, 300 mM NaCl, 5 mM EDTA, and 0.1% Triton X-100 (wt/vol) and eluted in 2× Tris-Glycine SDS samples buffer containing 10% 2-mercaptoethanol.

**Mass spectrometry**. RFP pull-down proteins were separated by SDS-PAGE (4–20% gradient gels, Thermo-Fisher) and stained with coomassie blue R-250. Protein bands were excised and sequentially reduced with dithiothreitol and alkylated with iodoacetamide. Proteins were then digested with trypsin. The resulting peptide mixtures were analyzed with an LTQ Orbitrap Velos (Thermo Fisher Scientific) equipped with a nanoLC system (Eksigent). Peptide IDs were assigned with Mascot 2.3 (Matrix Science) and manually validated using Scaffold 3 software (Proteome Software). For label-free quantitation, peptide peak areas were calculated with Proteome Discoverer 1.3 (Thermo Fisher Scientific).

**PIP2 phosphatase activity assay**. COS-7 cells were seeded in fibronectin-coated glass-bottom 35 mm dishes (CellVis, Mountain View, CA) and transfected for 18 h with 0.1 µg PH-PLCD1-GFP, 0.3 µg Lyn$_{11}$-FRB-iRFP and either 0.6 µg mCherry-FKBP-INPP5E or 0.6 µg mRFP-FKBP-TMEM55b pre-complexed with 3 µg lipofectamine 2000 in 0.2 ml Opti-MEM. Cells were imaged in 1.6 ml FluoroBrite media (Life technologies, Thermo-Fisher/Gibco, Carlsbad, CA, USA) supplemented with Glutamax and 10% FBS using a ×100 1.45 NA oil-immersion objective on a Nikon A1R laser scanning confocal microscope, imaging multiple positions in the dish every 30 s. After 2 min baseline imaging, rapamycin was added to 1 µM to induce translocation of the FKBP-chimeras to the plasma membrane. For analysis, the iRFP plasma membrane signal was used to generate a binary mask to quantify the normalized fluorescence intensity in the GFP and mRFP/mCherry channels over time as previously shown[65].

**Malachite green phosphatase assay**. TMEM55A and TMEM55B CD were cloned in pGEX-5X-1 expression vectors, transformed into Rosetta BL21(DE3) pLys strains and grown to an OD600 of ~0.4, before inducing protein expression with 100 µM IPTG at room temperature overnight. Bacteria were harvested, lysed using Bug Buster (EMD Millipore) and purified on glutathione agarose, before eluting with reduced glutathione. Finally, protein was buffer exchanged on Dowex columns into KHME buffer (110 mM potassium acetate, 20 mM HEPES, 2 mM magnesium chloride, 100 µM EGTA, PH 7.0). Aliquots were flash-frozen in liquid nitrogen and stored at –80 °C before use. Purified His$_6$-tagged SHIP2 was purchased from Echelon Biosciences. In total, 250 ng of each protein in KHME was mixed with an equal volume of diC$_8$-PtdIns-4,5-P$_2$ (Echelon Biosciences) to reach a final concentration of 20 µM in 50 µl. Reactions were terminated after 15 min at 37 °C by the addition of 100 µl Malachite Green reagent (Biomol). After 20 min for the color to develop, absorbance was read at 620 nm. A standard curve of 31–2000 pmol phosphate was prepared for each experiment. For each measurement, blank measurements (20 µM substrate with no enzyme, as well as 250 ng protein with no substrate) were subtracted.

**Subcellular fractionation**. Hela cells infected with adenovirus expressing TMEM55B-GFP or null adenovirus were harvested and lysed in hypotonic buffer (20 mM Tris-HCl pH 7.5, 250 mM sucrose, 5 mM Mg$_2$Cl, 1 mM EDTA and 1 mM DTT) containing protease and phosphatase inhibitors by 30 passages through a 28 G needle. Lysates were centrifuged at $16,000 \times g$ for 10 min at 4 °C to obtain the postnuclear supernatant (PNS). PNS were then centrifuged at $100,000 \times g$ for 1 h at 4 °C to obtain supernatant (S100) and pellet (P100) fractions. P100 were resuspended in Laemmli buffer to a volume equal to the volume of S100. Equal volumes of all fractions were analyzed by SDS-PAGE (4–20% gradient gels, Thermo-Fisher) and immunoblotting.

**Statistical analysis**. Obtained data were processed in Excel (Microsoft Corporation) to generate bar charts and perform statistical analyses. Pairwise post tests or one-way ANOVA were run for each dependent variable, as specified in each figure legend. To assess statistical significance between different cumulative intensity distribution profiles we calculated $P$ values by using prism software. Average distribution curves were individually fit to a non-linear regression based on the Michaelis-Menten equation using Prism 7 software. $P$ values were calculated from extra sum-of-squares $F$ test. $P \leq 0.05$ was considered statistically significant (*) and

$P \leq 0.01$ (**), $P \leq 0.001$(***), $P \leq 0.0001$(****) extremely significant. $P > 0.05$ was considered not significant (ns).

**Data availability**. The authors declare that the data supporting the findings of this study are available within the paper and its Supplementary Information Files.

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

## Acknowledgements

This work was supported by the Intramural Research Program of the National Institutes of Health, National Heart, Lung, and Blood Institute (NHLBI). We thank Dr. Medhi Pirooznia, director of the NHLBI Bioinformatics and Computational Core Facility, for assistance and advice on statistical analysis.

## Author contributions

R.W. and J.A.M. were involved in the experimental strategy, performed the experiments, analyzed the data, and participated in the preparation of the manuscript; J.P., Ra.W. and G.R.V.H. performed the PIP2 phosphatase activity assays and provided valuable advice; R.P. designed the research, analyzed the data, supervised the project, and wrote the manuscript. All the authors reviewed the manuscript.

## Additional information

**Competing interests:** The authors declare no competing financial interests.

