## [Peer Review File · Nature Communications]

Reviewers' comments:

Reviewer #1 (Remarks to the Author):

The manuscript by Willett and colleagues has been properly revised so that it could answer most of the comments raised by reviewers satisfactory; thus I would recommend publication.

Reviewer #2 (Remarks to the Author):

This revised manuscript reports a study identifying a role for the lysosomal integral membrane protein TMEM55B its binding partner, JIP4 in dynein-dependent transport of lysosomes towards the minus end of microtubules in the perinuclear region. The original manuscript showed that TMEM55B is upregulated by the transcription factors TFEB/3 in response to starvation and that the TFEB/TMEM55B/JIP4 pathway is involved in mediating nutrient-dependent transport of lysosomes to the perinuclear region to stimulate efficient autophagic flux. The revised manuscript has been strengthened by modifications in response to the previous reviewers' comments and by the addition of further data supporting a role for alterations in the TMEM55B/JIP4 pathway contributing to the pathology of Niemann-Pick type C disease as well as new data showing a role for the pathway in the cellular response to acute stress and suggesting that the pathway links cellular signalling to lysosomal distribution.

Overall, the data presented is technically sound and there is strong evidence for the conclusions drawn. The results are novel and the paper will be of interest to cell biologists, in particular those studying intracellular membrane traffic, the endocytic pathway, autophagy and lysosome function, as well as biomedical scientists interested in a broad range of lysosomal diseases. The authors have provided comprehensive and appropriate responses to the reviews of the previous version of the manuscript.

Reviewer #3 (Remarks to the Author):

I revised the new version of the article and the point-by-point response by Willett et al. I have few remaining questions that can be addressed experimentally and that, in my opinion, are needed to strength authors conclusions:

1) Figure S9b: The experiment showing variations in TMEM55B protein levels should be repeated using cells that lack TFEB and/or TFE3. This would demonstrate that the starvation-dependent TMEM55B increase is indeed due to transcriptional regulation.

2) Figures 6K,L and S9D: Can the authors repeat these experiments loading control and silenced samples on the same gel? I am not sure that samples loaded on different gels can be compared. Currently, the results are consistent with a defective AV biogenesis rather than a defective AV-Lys fusion upon TMEM55B and JIP4 silencing.

3) In the point by point response letter (point 1), the authors state that two hours of starvation are sufficient to observe the transcriptional-regulated perinuclear clustering of lysosomes. In the paper, they show that treatment with curcumin for two hours leads to the redistribution of lysosomes towards the central part of the cell (figure 8). However, in this case the authors postulate that curcumin-mediated lysosomal movement is too fast and hence mediated by a post-

translational mechanism. Can the authors discuss this apparent discrepancy? Why have they not considered transcriptional regulation as possible mechanism to explain the results obtained with curcumin? In addition, what happens to P38 during starvation and to TFEB during curcumin treatment? In other words why can't the two regulatory processes coexist and co-operate?

Reviewer #1 (Remarks to the Author):

The manuscript by Willett and colleagues has been properly revised so that it could answer most of the comments raised by reviewers satisfactory; thus I would recommend publication.

We thank the reviewer for the support

Reviewer #2 (Remarks to the Author):

This revised manuscript reports a study identifying a role for the lysosomal integral membrane protein TMEM55B its binding partner, JIP4 in dynein-dependent transport of lysosomes towards the minus end of microtubules in the perinuclear region. The original manuscript showed that TMEM55B is upregulated by the transcription factors TFEB/3 in response to starvation and that the TFEB/TMEM55B/JIP4 pathway is involved in mediating nutrient-dependent transport of lysosomes to the perinuclear region to stimulate efficient autophagic flux. The revised manuscript has been strengthened by modifications in response to the previous reviewers' comments and by the addition of further data supporting a role for alterations in the TMEM55B/JIP4 pathway contributing to the pathology of Niemann-Pick type C disease as well as new data showing a role for the pathway in the cellular response to acute stress and suggesting that the pathway links cellular signalling to lysosomal distribution.

Overall, the data presented is technically sound and there is strong evidence for the conclusions drawn. The results are novel and the paper will be of interest to cell biologists, in particular those studying intracellular membrane traffic, the endocytic pathway, autophagy and lysosome function, as well as biomedical scientists interested in a broad range of lysosomal diseases. The authors have provided comprehensive and appropriate responses to the reviews of the previous version of the manuscript.

We thank the reviewer for the support

Reviewer #3 (Remarks to the Author):

I revised the new version of the article and the point-by-point response by Willett et al. I have few remaining questions that can be addressed experimentally and that, in my opinion, are needed to strength authors conclusions:

1) Figure S9b: The experiment showing variations in TMEM55B protein levels should be

repeated using cells that lack TFEB and/or TFE3. This would demonstrate that the starvation-dependent TMEM55B increase is indeed due to transcriptional regulation.

RESPONSE: Unfortunately, our antibody does not recognize endogenous TMEM55B in mouse, so we were unable to compare TMEM55B protein levels between control and TFEB/TFE3 DKO MEFs (please note that in **Figure 6f** we show that transcriptional upregulation of TMEM55B in response to starvation is inhibited in TFEB/TFE3-DKO MEFs). In order to address the point raised by the reviewer, we compared TMEM55B protein levels between shcontrol and shTFEB-treated HeLa cells. We found TMEM55B protein levels were not only decreased in TFEB-depleted cells under basal conditions but failed to increase (and even got reduced) following starvation. In contrast, a modest but consistent increase in TMEM55B protein level was observed in starved control cells. These results are now shown in **Figure I**, attached to this rebuttal letter.

2) Figures 6K, L and S9D: Can the authors repeat these experiments loading control and silenced samples on the same gel? I am not sure that samples loaded on different gels can be compared. Currently, the results are consistent with a defective AV biogenesis rather than a defective AV-Lys fusion upon TMEM55B and JIP4 silencing.

RESPONSE: Please note that the samples shown in **Supplementary Figure 9d** were originally loaded in the same gel (see film scan in **Figure IIa**, attached to the rebuttal letter).

Unfortunately, we did not have enough material left to re-run the samples shown in **Figure 6k** in the same gel. For this reason, we now show the samples corresponding to another of the three independent experiments that were used for the quantification shown in **Figure 6i**. These samples were originally run in the same gel and are now shown in **Figure IIb** (attached to this rebuttal letter) and **Figure 6k**. Please note that we still show the samples in different panels in Figure 6k and Supplementary Figure 9d due to space limitations; however, we now mention that the samples were run in the same gel in the corresponding Figure Legends.

3) In the point by point response letter (point 1), the authors state that two hours of starvation are sufficient to observe the transcriptional-regulated perinuclear clustering of lysosomes. In the paper, they show that treatment with curcumin for two hours leads to the redistribution of lysosomes towards the central part of the cell (figure 8). However, in this case the authors postulate that curcumin-mediated lysosomal movement is too fast and hence mediated by a post-translational mechanism. Can the authors discuss this apparent discrepancy? Why have they not considered transcriptional regulation as possible mechanism to explain the results

obtained with curcumin? In addition, what happens to P38 during starvation and to TFEB during curcumin treatment? In other words why can't the two regulatory processes coexist and cooperate?

RESPONSE: As suggested by the reviewer, we have now assessed whether TMEM55B transcriptional upregulation may be the reason of the robust lysosomal clustering observed upon curcumin treatment. For this, control MEFs were treated with curcumin for different periods of time and TMEM55B mRNA levels were measured by qRT-PCR. As seen in **Figure IIIa** (attached to this rebuttal letter), TMEM55B mRNA levels did not increase in response to curcumin. In contrast, we observed a robust increase in TMEM55B mRNA levels in response to starvation. These results suggest that the initial lysosomal clustering in response to curcumin does not require TMEM55B transcriptional upregulation.

We also measured p38 activation in response to starvation. Whereas we observed a slight p38 activation in cell starved for 4h (1.3-fold increase when compared with untreated cells), the activation of p38 in response to curcumin was much more robust (2.5- and 3.3-fold increase upon 4 or 6h incubation with curcumin, respectively) (**Figure IIIb**, attached to this rebuttal letter). Future studies will further address the possibility that TMEM55B transcriptional and post-translational regulation coexist and cooperate to accurately modulate lysosomal positioning in response to a variety of stress conditions.

a**b**
Supporting Figure I: (a) Immunoblot of lysates from HeLa shControl or shTFEB cells starved in EBSS for the indicated times. (b) Quantification of TMEM55B protein levels. TMEM55B/Actin ratios increase in response to starvation in shControl cells while protein levels decrease in shTFEB cells.

a**b**
Supporting Figure II: (a) Scan of Western blot exposed film corresponding to Supplemental Figure 9d. Representative immunoblot of HeLa cells treated with Control or TMEM55B siRNA. Lysates were processed on the same gel/membrane (b) Scan of Western blot exposed film corresponding to Figure 6k. Representative immunoblot of HeLa cells treated with Control or JIP4 siRNA. Lysates were processed on the same gel/membrane.

a**b**
Supporting Figure III: (a) Relative quantitative real-time PCR analysis of TMEM55B mRNA transcript levels normalized to GAPDH in MEF cells treated with 20 μ M curcumin for the times indicated or starved for 4 h in EBSS. Curcumin treatment does not initiate increased expression of TMEM55B as seen with starvation. **(b)** Immunoblot of lysates from MEF cells treated with 20 μ M curcumin for the times indicated or starved for 4 h in EBSS. Curcumin treated cells have an increase in p38 phosphorylation compared to starved or untreated cells.

REVIEWERS' COMMENTS:

Reviewer #3 (Remarks to the Author):

I have no further comments and I recommend publication of the study.